# In situ dissection of domain boundaries affect genome topology and gene transcription in *Drosophila*

Rodrigo G. Arzate-Mejía[1], Angel Josué Cerecedo-Castillo[1], Georgina Guerrero[1], Mayra Furlan-Magaril[1,2 ✉] & Félix Recillas-Targa[1,2 ✉]

Chromosomes are organized into high-frequency chromatin interaction domains called topologically associating domains (TADs), which are separated from each other by domain boundaries. The molecular mechanisms responsible for TAD formation are not yet fully understood. In *Drosophila*, it has been proposed that transcription is fundamental for TAD organization while the participation of genetic sequences bound by architectural proteins (APs) remains controversial. Here, we investigate the contribution of domain boundaries to TAD organization and the regulation of gene expression at the *Notch* gene locus in *Drosophila*. We find that deletion of domain boundaries results in TAD fusion and long-range topological defects that are accompanied by loss of APs and RNA Pol II chromatin binding as well as defects in transcription. Together, our results provide compelling evidence of the contribution of discrete genetic sequences bound by APs and RNA Pol II in the partition of the genome into TADs and in the regulation of gene expression in *Drosophila*.

---

[1] Departamento de Genética Molecular, Instituto de Fisiología Celular, Universidad Nacional Autónoma de México, 04510 Ciudad de México, México. [2]These authors jointly supervised this work: Mayra Furlan-Magaril, Félix Recillas-Targa. ✉email: mfurlan@ifc.unam.mx; frecilla@ifc.unam.mx

In eukaryotes, the genome is non-randomly folded in the nuclear space[1,2]. At the sub-megabase scale, chromosomes are organized into high-frequency chromatin interaction domains termed topologically associating domains (TADs) separated from each other by domain boundaries[3–6]. The presence of TADs has been described in different species suggesting that they represent a common feature of genome organization[4–7]. From a transcriptional perspective, TADs can facilitate the establishment of specific regulatory landscapes by bringing into proximity enhancers and promoters, while at the same time, precluding unspecific regulatory communication with sequences outside the domains[8,9]. Disruption of TAD organization by genetic manipulation of domain boundaries can result in ectopic regulatory communication of sequences leading to gene misexpression and significant consequences in cell physiology and organismal development[10–13].

TAD boundaries in mammals are occupied by CTCF and the Cohesin complex[6,14]. Manipulation of the CTCF binding motif at domain boundaries or the acute degradation of CTCF or subunits of the Cohesin complex leads to loss of TADs and alterations in gene expression[3,15–18]. In Drosophila, TAD boundaries are detected at chromatin accessible regions enriched for active histone marks and occupied by multiple APs including CP190, BEAF-32, M1BP, Su(Hw), and CTCF among others[19–21]. Analysis of high-resolution Hi–C datasets suggests that in the fly, genome domains emerge as a consequence of the transcriptional activity of sequences within the domain while APs modulate interactions between domains[22]. Genome-wide inhibition of transcription in embryos and cell lines results in a significant decrease in TAD boundary insulation; however, TADs do not disappear[22–24]. On the other hand, depletion of BEAF-32, an AP enriched at TAD boundaries, results in minor changes in genome organization[19]. Then, whether TADs in Drosophila are the result of self-aggregation of chromatin regions with similar transcriptional states or emerge as a consequence of the activity of discrete genetic elements bound by APs acting as chromatin insulators remains controversial.

The Notch gene locus in Drosophila provided one of the first evidence linking chromatin insulation with domain formation and gene expression[25,26]. Deletion of a ~900-bp region upstream of the Notch promoter in the mutant allele fa(swb) results in loss of the interband 3C6-7 and fusion of the band containing Notch with the upstream band harboring the kirre locus[26]. This cytological effect leads to a reduction in Notch expression and an eye phenotype in adult mutant flies[26]. In Drosophila, the bands observed in polytene chromosomes correspond to TADs, and inter-band regions have a close correspondence with TAD boundaries[27,28], which suggest that the Notch locus is a TAD itself and the 5′ intergenic region is a domain boundary.

Here, we study the contribution of domain boundaries to TAD organization and the regulation of gene expression at the Notch locus in Drosophila. We show that Notch is organized into two TADs with boundaries overlapping the 5′ intergenic region and an intronic enhancer. We generated CRISPR-Cas9 mediated deletions of the Notch domain boundaries and evaluated their topological effect by in nucleus Hi–C. Consistent with cytological data, removal of the entire 5′ intergenic region results in the fusion of the first domain of Notch with the upstream TAD. Furthermore, we uncovered that portions of the 5′ intergenic region, with binding sites for specific APs, act as discrete chromatin insulators between the upstream gene kirre and Notch, with removal of all regions necessary for TAD fusion. In all cases, topological disorganization of the first domain of Notch is accompanied by loss of APs and RNA Pol II occupancy at the disrupted boundary, reduction in transcription of the exons that reside within the affected domain and changes in the transcription levels of genes located in adjacent TADs. Removal of the intragenic enhancer resulted in the fusion of the two Notch domains and decreased transcription along the locus. Moreover, in wild-type cells, this genomic element is also the anchor of a megabase-sized domain, which is lost upon enhancer deletion, and results in changes in transcription of the genes located inside the domain. Together, our data provide compelling evidence on the contribution of domain boundaries to TAD formation and the control of gene expression in Drosophila.

## Results

***Notch* is organized into two topological domains in S2R+ cells.** To characterize the topological organization of Notch, we performed in nucleus Hi–C using a 4-cutter restriction enzyme (MboI) in the embryonic cell line S2R+ in triplicate (see Methods)[29] reaching a minimum of 89% valid pairs per replicate (Supplementary Table 1). The high correlation between replicates allowed us to merge them into a single dataset (Supplementary Fig. 1a). A heatmap of Hi–C data binned at 1 kb resolution shows that Notch is organized into two topological domains in S2R+ cells (Fig. 1a), consistent with Hi–C data from early embryos[28] and with high-resolution in situ hybridization analysis of the Notch locus in polytene chromosomes[30]. Identification of TADs genome-wide also shows that Notch is partitioned into two TADs in all Hi–C replicates (Supplementary Fig. 1b).

The two topological domains at Notch split the gene unevenly. Domain 1 (D1; size 31 kb) comprises from the start of the gene to exon six while Domain 2 (D2; size 9 kb) contains the remaining 4 exons (Fig. 1a). The Notch locus is located between kirre and dnc genes. kirre is a ~400-kb transcriptional unit with multiple isoforms and is partitioned into two TADs of ~200 kb each (termed kirre domain-1 and kirre domain-2) while the dnc locus is contained within a ~150-kb TAD (termed dnc domain). kirre domain-2 is adjacent to the 5′ end of Notch while the dnc domain is next to the 3′ end of the locus (Supplementary Fig. 1c).

We detected two boundaries spanning the Notch locus (Fig. 1a). The first boundary (termed B1) is located at the intergenic region between kirre and Notch genes and separates the D1 domain of Notch from the upstream TAD (kirre domain-2). The second boundary (termed B2) is located at intron five and the start of exon six and demarcates the transition between D1 and D2 domains of Notch (Fig. 1a). In Drosophila, domain boundaries are chromatin accessible sites occupied by APs and components of the transcriptional machinery, like RNA Pol II[19,21,24]. Analysis of public ChIP-seq data against APs in S2/S2R+ cells reveals that the B1 boundary is a high occupancy Architectural Protein Binding Site (APBS) bound by more than 8 APs and RNA Pol II (Fig. 1a)[21]. However, motif analysis revealed just the presence of CTCF and M1BP binding motifs (p-value < 0.0001) (Fig. 1d). The B2 boundary overlaps an enhancer at intron five described to control Notch transcription during larval development and an RNA Pol II enriched region at exon 6 (ref. [31]) (Fig. 1a). The enhancer is a low occupancy APBS highly enriched for histone H3K4me1 while exon 6 shows prominent enrichment for histone H3K4me3 and binding of RNA Pol II. The presence of RNA Pol II at the B2 boundary suggests that the D2 domain could reflect an independent transcriptional unit of Notch resembling a mini-gene domain[22]. In support of this, a GenBank entry for a cDNA sequence from Drosophila embryos (BT023499.1) matches the sequence between exon 6 to the end of the Notch gene (Supplementary Fig. 1d).

We validated the binding of CTCF, RNA Pol II (pSer2 and pSer5), and the enrichment of histone marks H3K4me3, H3K27ac, and H3K27me3 at Notch domain boundaries by qChIP. Consistent with ChIP-seq data, CTCF binding is only

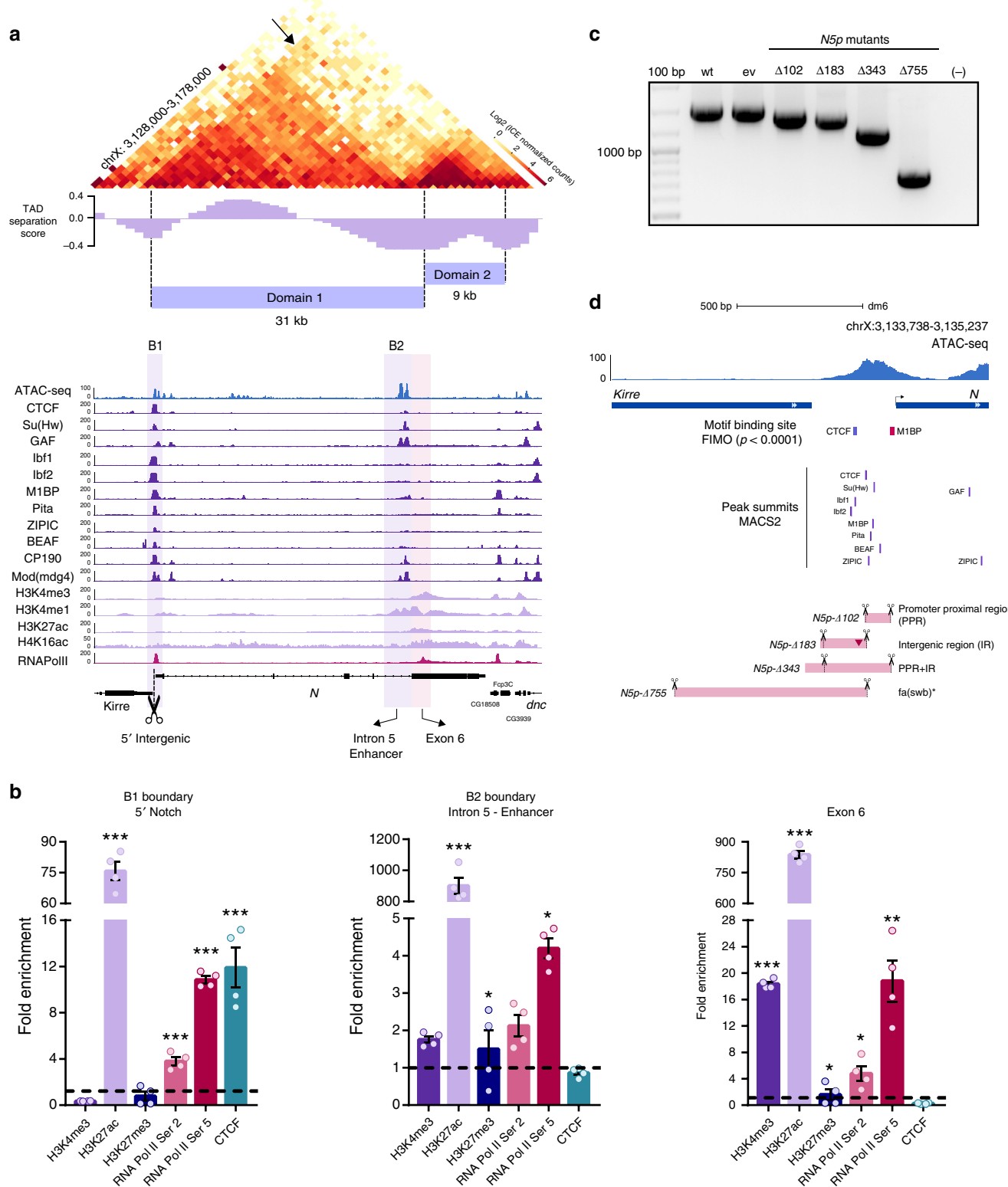

detected at the B1 boundary while the H3K27ac histone mark is ~10 times more enriched at the enhancer and exon 6 of the B2 boundary than at the B1 boundary (Fig. 1b). In contrast, the RNA Pol II (both pSer2 and pSer5) occupies both boundaries. Interestingly, exon 6 shows the highest enrichment of RNA Pol II and histone H3K4me3 (Fig. 1b).

In *Drosophila*, TADs first emerge around the onset of Zygotic Genome Activation (ZGA)[24,32]. To get insight into the dynamics

of domain formation at the *Notch* locus during early development, we analyzed public *in nucleus* Hi–C data[24]. A domain boundary at the 5′ intergenic region of *Notch* (B1) is detected at nuclear cycle 13 (nc13) before ZGA while a second domain boundary (B2) between intron five and exon six is detected at nc14 concomitant with ZGA. Both boundaries partition the *Notch* locus into two domains (Supplementary Fig. 2a) as observed in the S2R+ cell line. The establishment of TADs

**Fig. 1 The 3D organization of *Notch* in S2R+cells. a** Hi-C heatmap at 1-kb resolution covering a 50-kb region centered in *Notch*. TAD separation score for the locus is shown. The *Notch* locus is partitioned into two topological domains depicted as Domain 1 and Domain 2. The B1 boundary interacts with the full *Notch* locus (see arrow). Public ChIP-seq data for Architectural Proteins (APs), RNA Pol II, and histone marks for S2/S2R+ cells is shown below the heatmap. The position of the domain boundaries relative to *Notch* is highlighted and depicted as B1 and B2. **b** qChIP experiments in wild-type S2R+ cells using different antibodies for the 5′ intergenic region (left), the intronic enhancer (center), and exon 6 (right). Significant differences in enrichment against IgG were calculated using a *t*-test. *p*-value *$p < 0.05$, **$p < 0.01$, ***$p < 0.001$. Error bars represent the standard error of the mean (s.e.m.) of four replicates ($n = 4$). Source data are provided as Source Data File. **c** Genotypification of CRISPR mutant clones with deletions spanning the B1 boundary. Wild-type expected amplicon size, 1500 base pairs (bp). ev, empty vector. The number associated with each CRISPR mutant clone represents the number of base pairs deleted. Source data are provided as Source Data File. **d** Schematic representation of CRISPR mutant clones for the B1 boundary. Pink rectangles represent the deleted regions. Scissors indicate the position of sgRNAs used for CRISPR mediated genome editing. The red inverted triangle represents a 28-bp insertion in the mutant *5pN-Δ183*. *fa(swb)** indicates that the mutant *5pN-Δ755* is similar to the *fa(swb)* allele[26]. Motif binding sites for APs detected by FIMO (*p*-value < 0.0001) are shown as boxes for CTCF and M1BP as well as peak summits for DNA-binding APs shown in **a**.

during embryonic development co-occurs with de novo recruitment of RNA Pol II and gain of chromatin accessibility genome-wide[24]. We analyzed public data for ATAC-seq[33] and ChIP-seq for RNA Pol II binding[34], TATA-binding protein (TBP)[35], and Zelda[36]. We found that the establishment of boundaries at *Notch* strongly correlates with the progressive gain of chromatin accessibility and the binding of Zelda up to nc14 (Supplementary Fig. 2b).

In *Drosophila*, dosage compensation occurs at active loci of the male X chromosome due to the recruitment of the MSL complex which results in increased transcription and the deposition of the histone post-translational modification H4K16ac into the gene bodies of dosage compensated loci[37,38]. Recent observations suggest the existence of sex-specific differences in genome organization between the X chromosomes of female and male cells, which correlate with differences in the binding of APs at domain boundaries and with the differential enrichment of H4K16ac at dosage compensated loci[39,40]. The *Notch* locus is located in the X chromosome and shows differential enrichment of H4K16ac as well as different expression level between male and female suggesting is dosage compensated (Supplementary Fig. 3a). To investigate differences in genome organization at the *Notch* locus between female and male cells we re-analyzed Hi–C datasets derived from the female embryonic cell line Kc167 (refs. [23,41]) and the male CNS-L3 cell line BG3 (ref. [42]) and compared *Notch* 3D organization with the one observed in the male embryonic cell line S2R+. We observe two domains spanning the *Notch* locus in the male derived cell lines (S2R+ and BG3) in contrast with a single domain organization of *Notch* in the female derived cell line Kc167 (Supplementary Fig. 3a). To relate the observed topological differences at *Notch* with the presence of the histone post-translational modification H4K16ac and the binding of APs we obtained processed signal files from modENCODE for H4K16ac and re-analyzed publicly available ChIP-seq data sets of APs for the Kc167 cell line[23,41,43]. We observe a clear difference in the enrichment of H4K16ac between the female cell line (Kc167) and the male cell lines (S2R+ and BG3), with H4K16ac being highly enriched at the *Notch* region encompassing the *D2* domain detected in male cells (exon 6–exon 9) while the histone post-translational modification H3K27me3 is enriched at the same region in female cells and overall depleted in male cells (Supplementary Fig. 3a). Furthermore, the two domain organization of *Notch* and the enrichment of H4K16ac at D2 Domain in male cells correlate with higher expression levels of *Notch* when compared to a female derived cell line (Supplementary Fig. 3a). We also observe differences in APs occupancy at the *Notch* locus between female and male cell types which correlates with the presence of domain boundaries along the locus (Supplementary Fig. 3b, c). In particular, while the B1 at the 5′ end of *Notch* is observed in both female and male derived cell lines and this correlates with the binding of multiple APs, CP190,

and RNA Pol II, the genomic region encompassing the B2 boundary detected in male cells, shows an overall reduction in the binding of APs in the female derived Kc167 cell line (Supplementary Fig. 3c). Importantly, the observed difference in APs binding at the B2 boundary genomic region in Kc167 cells is not due to differential chromatin accessibility since Kc167 and S2 cells show a remarkable similar ATAC-seq profile at the *Notch* locus[44] (Supplementary Fig. 3b, c). The dynamic organization of this locus contrast with early observations suggesting a mostly invariant organization of the genome in *Drosophila* and supports recent observations that boundaries are dynamic between cell types and that these variability correlates with the binding of APs[39,42].

Together, these results indicate that the *Notch* locus is organized into two topological domains that isolate the gene from neighboring TADs in S2R+ cells. *Notch* domain boundaries are enriched for active histone marks and occupied by RNA Pol II and for a variable number of Architectural Proteins. During embryonic development, domain boundaries are detected before transcription of the locus and strongly correlate with the progressive acquisition of chromatin accessibility and RNA Pol II binding. Also, the *Notch* locus shows sex-specific topological organization which correlates with differences in transcription, the enrichment of histone modifications like H4K16ac and the binding of APs.

**The 5′ intergenic region of *Notch* is a chromatin insulator**. To directly evaluate the chromatin insulator activity of the B1 boundary of *Notch*, we generated CRISPR-Cas9 mediated deletions in S2R+ cells (Fig. 1c, d; Supplementary Fig. 4a). We isolated mutant clones for four different deletions and named them based on the number of base pairs deleted (Supplementary Fig. 4b, c). To assess the topological effects of the mutant clones, we performed *in nucleus* Hi-C in duplicates and visualized topological effects by plotting heatmaps of the valid Hi–C pairs at different resolutions and over different genomic segments. We also generated Virtual-4C profiles using Hi–C data from wild-type and mutant clones to visualize the interactions profile from different viewpoints along the Notch locus.

Overall, the *in nucleus* Hi–C libraries generated for mutant clones are of good quality reaching at least 89% valid pairs and replicates are highly correlated (Pearson correlation coefficient 0.91–0.94). Also, the correlation between the different mutants and the wild-type samples is high (0.89–93) which suggest only local topological effects upon deletion of small DNA fragments (<800 bp), as expected (Supplementary Fig. 5). Normalized heatmaps at 1-kb resolution centered at *Notch* reveal topological effects in all mutants (Fig. 2a). Deletion of the Promoter Proximal Region of *Notch* (PPR; *5pN-Δ102*) removes M1BP DNA-binding site but not CTCF DNA-binding site as well as the ChIP-seq peak

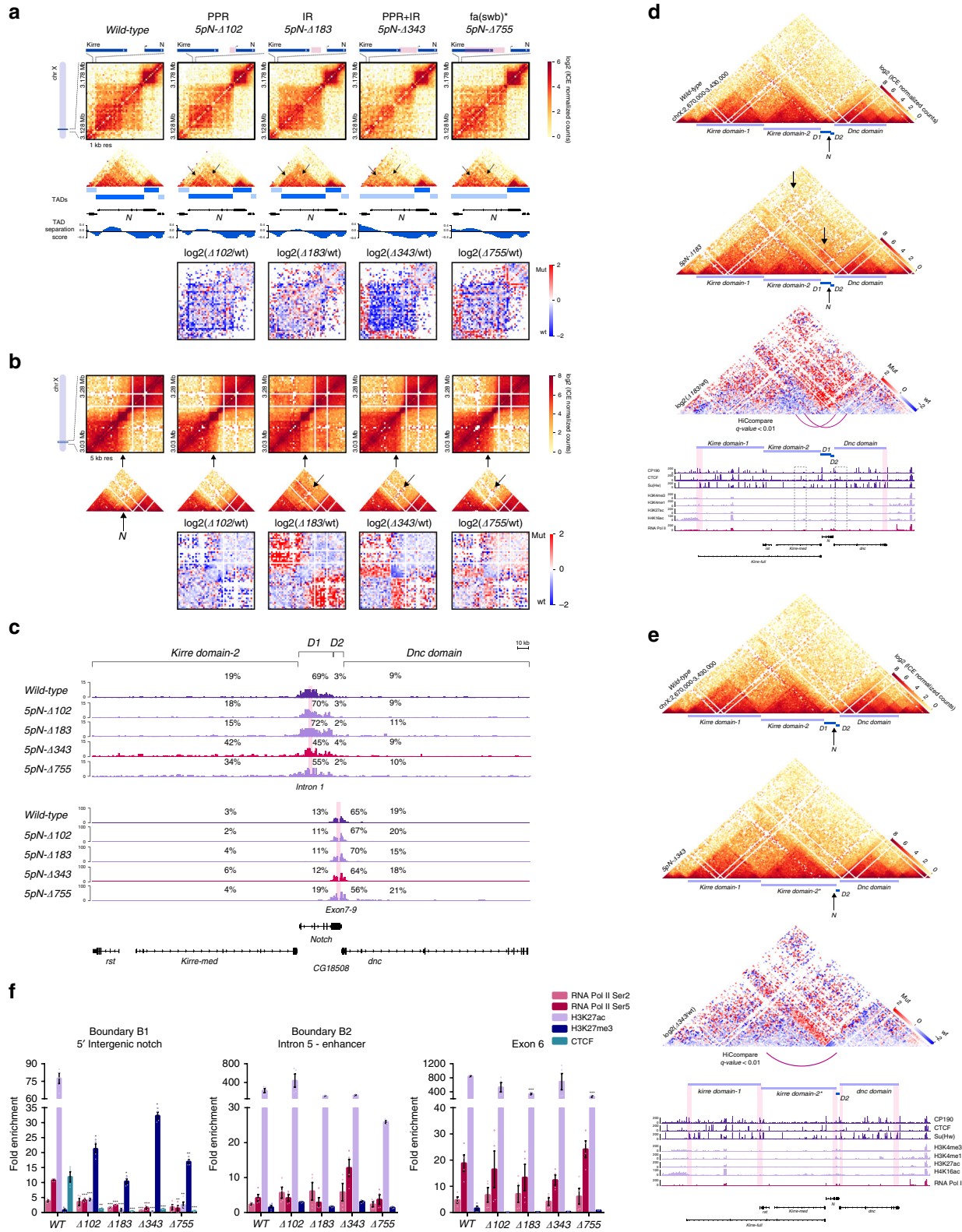

summits of CTCF, Su(Hw), Pita and BEAF-32 (Fig. 1d) and results in a significant loss of intra-TAD interactions inside the *D1* domain (p-value 6.87e-9, Supplementary Fig. 6a), particularly between its two subdomains, while the *D2* domain shows a moderate, although, no significant decrease in interactions (Fig. 2a; Supplementary Fig. 6a). The interaction profile of the 5′UTR of *Notch*, as evaluated by Virtual-4C, shows a decrease in interactions with the *D1* domain concomitant with a gain of ectopic interactions with the upstream *kirre domain-2* (Supplementary Fig. 6e). However, this trend is not observed when additional viewpoints within the *D1* domain are analyzed (Fig. 2c). Despite observed topological effects, the removal of the PPR is not sufficient to induce the fusion of the *D1* domain and the upstream *kirre domain-2*.

**Fig. 2 Deletion of the 5′ intergenic region of *Notch* results in TAD fusion. a** Top, Hi–C normalized heatmaps at 1-kb resolution covering a 50-kb region centered in *Notch*. Middle, triangular representation of Hi–C heatmaps, along with TADs and TAD separation score for each dataset. Arrows indicate regions with differences in genomic interactions. Bottom, Hi–C heatmaps of the log2 differences in interaction frequency between wild-type and the CRISPR mutants. PPR proximal promoter region, IR intergenic region. *fa(swb)** indicates that the mutant *5pN-Δ755* is similar to the *fa(swb)* allele[26]. **b** Top, Hi–C normalized heatmaps at 5-kb resolution covering a 250-kb region centered in *Notch*. The black arrow at the bottom of the plot indicates the position of the *Notch* locus. Middle, triangular representation of Hi–C heatmaps. Arrows indicate regions with differences in genomic interactions. Bottom, Hi–C heatmaps of the log2 differences in interaction frequency between wild-type and the CRISPR mutants. **c** Virtual-4C for wild-type and CRISPR mutant clones using viewpoints (pink rectangles) inside the D1 and D2 domain of *Notch*. Shown are the percent of interactions between the viewpoints and the *kirre domain-2*, the *Notch* D1 domain, the *Notch* D2 domain and the *dnc* domain. **d** Hi–C normalized heatmaps at 5-kb resolution for wild-type and the mutant *5pN-Δ183* covering a 760-kb region. Arrows indicate regions of gain of interactions in the mutant. A Hi–C heatmap of the log2 difference in interaction frequency between wild-type and the *5pN-Δ183* mutant is shown. Ectopic interactions detected in the mutant *5pN-Δ183* (arcs, dotted boxes) and ChIP-seq tracks for APs, histone marks and RNA Pol II are also shown. Note that the ectopic interactions between the *kirre* and *dnc* loci are constrained by the 5′ boundary of *kirre domain-1* and the 3′ boundary of *dnc* (highlighted in ChIP-seq track). **e** Hi–C normalized heatmaps at 5-kb resolution for wild-type and the mutant *5pN-Δ343* covering a 760-kb region. A Hi–C heatmap of the log2 difference in interaction frequency between wild-type and the *5pN-Δ343* mutant is shown. Ectopic interactions (arc) and ChIP-seq tracks for APs, histone marks and RNA Pol II are shown underneath. Highlighted regions correspond to domain boundaries. Note that deletion of the PPR + IR of *Notch* results in the fusion of *kirre domain-2* and the *Notch* D1 domain. **f** qChIP experiments in S2R + wild-type cells and CRISPR mutant clones. Fold enrichment was calculated against IgG. Significant differences in enrichment between wild-type and mutant clones were calculated using a *t*-test. *p*-value *$p < 0.05$, **$p < 0.01$, ***$p < 0.001$. Error bars represent the standard error of the mean (s.e.m.) of four replicates ($n = 4$). Source data are provided as Source Data File.

Deletion of the Intergenic Region (IR; *5pN-Δ183*) upstream of the PPR of *Notch* removes the CTCF DNA-binding site but not M1BP as well as the ChIP-seq peak summits of CTCF, Ibf1 and Ibf2 (Fig. 1d) and does not change the interaction frequency within the *Notch* locus nor results in inter-domain interactions with the *kirre domain-2* (Fig. 2a, c; Supplementary Fig. 6a, d). Then, this sequence does not directly promote domain formation or insulation of *Notch*. Instead, deletion of the IR resulted in a significant increase in interactions between the *kirre domain-2* and the *dnc domain* (*p*-value 2.2 e-16, Fig. 2b; Supplementary Fig. 6d), although ectopic interactions with the *dnc domain* extend up to the *kirre domain-1* (Fig. 2d; Supplementary Fig. 6c). The increase in inter-domain interactions is accompanied by a decrease in the number of interactions within each domain, likely reflecting the engagement of their sequences in ectopic contacts (Fig. 2d; Supplementary Fig. 6b). Moreover, visual inspection of Hi–C heatmaps at 5-kb resolution suggest that the 5′ and 3′ boundaries of the *kirre domain-1* and the *dnc domain*, respectively, engage in long-range interactions likely reflecting a preference of protein complexes at each boundary to physically interact (Fig. 2d). These boundaries are occupied by APs like CP190 and CTCF. Furthermore, a specific long-range interaction between the *kirre domain-2* and the *dnc domain* is also detected with anchors overlapping regions enriched for histone H4K16ac and APs (HiCcompare *q*-value < 0.01) (Fig. 2d). Therefore, the IR is essential to constrain interactions within the *kirre* locus while the PPR restrain interactions within the *D1* domain of *Notch*.

We hypothesized that deletion of both sequences, thus removing the DNA-binding motifs of M1BP and CTCF as well as all ChIP-seq peak summits for APs at the region, could result in the fusion of adjacent domains. Deletion of the PPR + IR region of *Notch* (*5pN-Δ343*) resulted in a dramatic loss of intra-domain interactions at the *D1* domain (*p*-value 2-2e-16) accompanied by a significant increase in interactions with the *kirre domain-2* (*p*-value 2.2 e-16) and therefore in TAD fusion (Fig. 2a–c, e; Supplementary Fig. 6a, d, e). Genome-wide identification of domain boundaries confirms the loss of the B1 boundary and identifies the enhancer at intron 5 of *Notch* as the new 3′ boundary of the *kirre domain-2* (Fig. 2a, e). Also, we observed a significant increase in interactions between the *kirre* and *dnc* loci (*p*-value 2-2e-16) and a decrease in the intra-domain interactions for the *D2* domain (*p*-value 0.01957) (Fig. 2a, b; Supplementary Fig. 6a–d). A similar topological effect is observed

with the *5pN-Δ755* mutant, although less severe, likely reflecting the presence of the M1BP binding motif and other APs (Fig. 2a–c; Supplementary Fig. 6a–e). Therefore, deletion of the B1 boundary results in loss of a chromatin insulator, fusion of the *D1* domain with the upstream TAD and a significant increase in contacts between the TADs surrounding *Notch*.

To relate topological effects with APs occupancy, we evaluated changes of CTCF and RNA Pol II binding at the B1 boundary region in all mutants by qChIP using mutant-specific primers (Supplementary Table 2). Deletion of the PPR (*5pN-Δ102*) leads to a marked decrease in the binding of RNA Pol II and CTCF, despite not removing the binding site for this protein (Fig. 2f). Deletion of the IR (*5pN-Δ183*) shows an even more pronounced decrease in RNA Pol II occupancy and loss of CTCF binding, consistent with the removal of its binding site (Fig. 2f). In agreement with topological effects, the mutant *5pN-Δ343* shows loss of binding for all evaluated proteins. Interestingly, the disruption of the B1 boundary in all mutants also results in a marked decrease in the histone post-translational modification H3K27ac accompanied by the gain of H3K27me3 (Fig. 2f). Importantly, the observed effects in H3K27ac, H3K27me3 and RNA Pol II are specific for the mutants of the B1 boundary region, as they enrichment remain mostly unaffected at the B2 boundary and exon 6 of *Notch* (Fig. 2f).

The marked decrease in RNA Pol II binding at the B1 boundary in the *5pN-Δ183* mutant suggest that sequences within the deleted region, including the CTCF binding site, are important for RNA Pol II recruitment. We evaluated the contribution of the CTCF DNA-binding motif in the recruitment of nuclear proteins by Electrophoretic Mobility Shift Assay (EMSA) using nuclear extracts of S2R+ cells. Mutation of the CTCF binding site results in a sharp decrease in shift and therefore in binding of nuclear proteins to that sequence, suggesting that the CTCF binding motif is necessary for the recruitment of regulatory proteins including RNA Pol II (Supplementary Fig. 6g).

Together, our results show that removal of a domain boundary results in TAD fusion. The *Notch* B1 boundary is a *bona fide* chromatin insulator with specific modules promoting effective insulation of neighboring regions of the genome. Importantly, topological effects after boundary removal correlate with changes in CTCF and RNA Pol II chromatin binding and H3K27ac and H3K27me3 deposition.

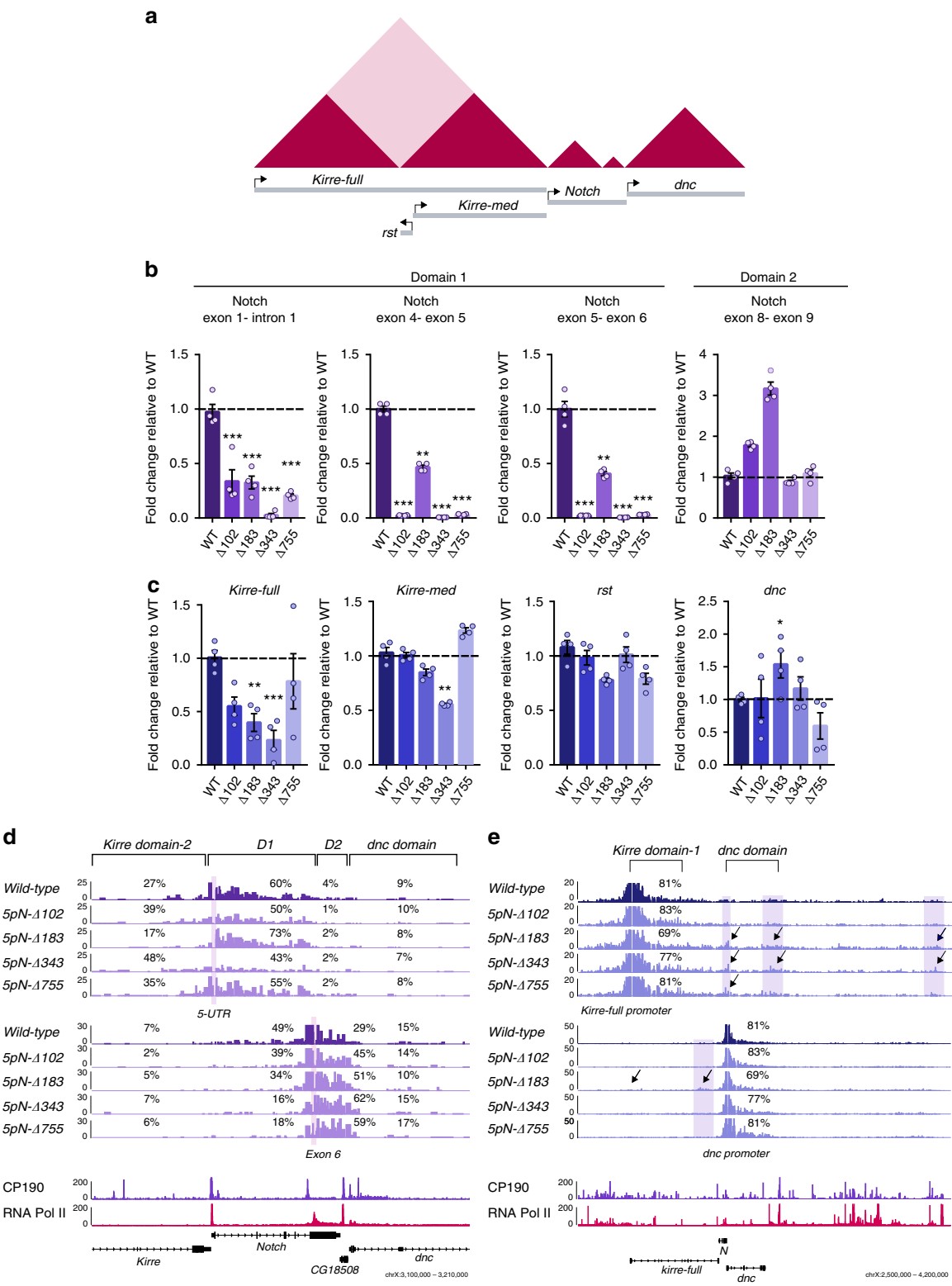

**Disruption of the 5′ insulator of *Notch* affects transcription**. To correlate the topological changes observed due to loss of the B1 boundary with gene expression, we measured transcription at the *Notch* locus by qPCR using primers spanning exon pairs at the *D1* and *D2* domains (Fig. 3a; Supplementary Table 2). We observe a reduction in transcription of exons within the *D1* domain for all mutants (Fig. 3b). Loss of transcription within the *D1* domain is consistent with loss of binding of RNA Pol II at the 5′ intergenic region of *Notch* (Fig. 2f) and correlate with both the loss of intra-

domain interactions at the *D1* domain (Fig. 2a; Supplemental Fig. 6a) and the gain of ectopic interactions of the 5′UTR, which contains the major TSSs of *Notch*, with *kirre domain-2* (Fig. 3d). In contrast, the transcript levels at the *D2* domain remain either unaffected (mutant *5pN-Δ343*) or show an increase (mutant *5pN-Δ183*) (Fig. 3b). In particular, the increased transcription at the *5pN-Δ183* correlates with a gain of interactions between sequences at the *D2* domain (Fig. 3d). The changes in transcription at the *D2* domain are also accompanied by a modest increase

**Fig. 3 Loss of the 5′ intergenic boundary of Notch affect gene expression. a** Schematic representation of the TAD landscape surrounding *Notch* and the genes tested for transcriptional changes. **b** Transcription of *Notch* as measured by RT-qPCR. Transcriptional quantifications in each CRISPR mutant compared to wild-type using three primer pairs spanning the *Notch* D1 domain and one pair for the *Notch* D2 domain. Significant differences between wild-type and CRISPR mutants were calculated using a *t*-test. $n = 3$, *p*-value *$p < 0.05$, **$p < 0.01$, ***$p < 0.001$. Error bars represent the standard error of the mean (s.e.m.) of four replicates ($n = 4$). Source data are provided as Source Data File. **c** Transcription of genes located at TADs flanking *Notch* as measured by RT-qPCR in wild-type and each CRISPR mutant. Significant differences between wild-type and CRISPR mutants were calculated using a *t*-test. $n = 3$, *p*-value *$p < 0.05$, **$p < 0.01$, ***$p < 0.001$. Error bars represent the standard error of the mean (s.e.m.) of four replicates ($n = 4$). Source data are provided as Source Data File. **d** Virtual-4C for wild-type and mutant clones using the 5′ UTR (top) and the exon 6 (bottom) as viewpoints. Shown are the percent of interactions between the viewpoints and regions within the *kirre domain-2*, the *Notch* D1 domain, the *Notch* D2 domain, and the *dnc* domain for both wt and all mutant samples. ChIP-seq tracks for CP190 and RNA Pol II are also shown. **e** Virtual-4C for wild-type and mutant clones using the promoter of the *kirre-full* isoform (top) and the promoter of *dnc* (bottom) as viewpoints. Shown are the percent of interactions between the viewpoints and the TADs in which they are located. Arrows indicate regions with ectopic interactions. ChIP-seq tracks for CP190 and RNA Pol II are also shown.

in RNA Pol II pSer2 occupancy at exon 6 (Fig. 2f). Therefore, disruption of the B1 boundary reduces transcription exclusively at the *D1* domain while the *D2* domain behaves as an independent topological and transcriptional unit likely due to the presence of the B2 boundary.

Deletions over the B1 border also result in ectopic long-range interactions between the topological domains flanking *Notch*. Therefore we evaluated the transcriptional level of *kirre-full*, *kirre-med*, and *rst* in all the B1 boundary mutants (Fig. 3a, c). We observe a ~50% reduction in the transcript levels of the *kirre-med* isoform just in the mutant *5pN-Δ343* while the *rst* gene does not show major changes in transcription (Fig. 3c). The transcriptional effect observed for *kirre-med* in the mutant *5pN-Δ343* could be related to the fusion of the *D1* domain and the *kirre-domain 2* due to loss of the B1 boundary. For the *kirre-full* transcript, we observe a reduction in the transcription levels of at least 50% for the mutant *5pN-Δ102* and up to 75% for the mutant *5pN-Δ343* (Fig. 3c). These changes in transcription are accompanied by a gain of ectopic long-range interactions between the promoter of the *kirre-full* gene locus and regions enriched in RNA Pol II and CP190 downstream of the locus (Fig. 3e). For example, in the mutant *5pN-Δ343* the *kirre-full* promoter region is engaged in ectopic interactions with the B2 boundary of *Notch* and a region >1 Mb away enriched for RNA Pol II and CP190 (Fig. 3e). Finally, we evaluated the transcript levels of the *dnc* gene located downstream of *Notch*. We observe a ~50% increase in transcription of *dnc* in the mutant *5pN-Δ183*, which is accompanied by a gain of ectopic interactions with the *kirre domain-2* (Fig. 3c, e). Therefore, disruption of *Notch* B1 domain boundary, is accompanied by both local and long-range transcriptional effects.

**Deletion of the intronic enhancer leads to fusion of Notch domains**. As shown, loss of the B1 boundary does not affect the topological organization of the *D2* domain of *Notch*, which suggests that it is an independent topological unit. The topological organization of the *D2* domain could result from the activity of the B2 boundary but could also reflect the self-association of the *D2* domain sequences due to transcription, histone modifications and RNA Pol II occupancy (Figs. 1a and 4a). To distinguish between these possibilities, we disrupted the B2 boundary by CRISPR-Cas9 mediated deletion of the enhancer element at intron 5 (Fig. 4a; Supplementary Fig. 7a) and evaluated the topological effects upon enhancer deletion by *in nucleus* Hi–C.

Removal of the enhancer sequence results in a striking increase in inter-domain interactions between both domains of *Notch* (*p*-value 2.2e-16, Fig. 4c) accompanied by a decrease in *D2* intra-domain interactions (*p*-value 0.0003939, Fig. 4c) and a gain of interactions with the *dnc domain* (Supplementary Fig. 7b, c). Virtual-4C using viewpoints over the *D1* and *D2* domains shows increase in inter-domain interactions, in particular, we observe a major gain in contacts between the region encompassing the

exons 7–9 of *D2* domain with the full *D1* domain, up to the B1 boundary (Fig. 4d). Genome-wide identification of domain boundaries confirms the loss of the B2 boundary and identifies a single TAD encompassing the *Notch* locus, a topological organization similar to the one observed in the female derived cell line Kc167 (Fig. 4b; Supplementary Fig. 3a).

Next, we evaluated RNA Pol II binding and H3K27ac and H3K27me3 enrichment in the enhancer mutant cell line. As expected, we observe a marked decrease of histone H3K27ac and RNA Pol II and a gain of H3K27me3 at the region surrounding the deleted enhancer (Fig. 4e). Unexpectedly, we also observe a significant loss of RNA Pol II binding and H3K27ac at exon 6 and at the B1 boundary but no changes in CTCF occupancy (Fig. 4e). This suggests that the enhancer is important for recruitment of RNA Pol II and deposition of histone H3K27ac at *Notch* regulatory elements.

The deletion of the enhancer and the loss of RNA Pol II at *Notch*, suggests there might be effects in the transcription of the locus. In contrast to what we observe upon B1 boundary removal, deletion of the B2 boundary resulted in downregulation of transcription along the locus (Fig. 4f). In particular, transcription within the *D2* domain is significantly affected, showing a ~75% reduction as compared to wild-type and in sharp contrast to the observed transcriptional effects upon B1 boundary deletion (Fig. 3b). In addition, transcription of the genes flanking *Notch* is also affected, and this correlates with the gain of ectopic interactions with regions enriched for RNA Pol II and CP190. For example, in the mutant cells, the promoter of the *kirre*-full isoform is engaged in long-range ectopic interactions with regions occupied by RNA Pol II and CP190 up to 1 Mb downstream of *Notch* (Fig. 4g). Together, our data show that upon enhancer loss contacts are gained between the two *Notch* domains. Also, it is essential to recruit RNA Pol II at *Notch* regulatory elements and therefore for *Notch* transcription.

**The intronic enhancer of Notch organizes a megabase-sized domain**. Visual inspection of wild-type Hi–C data binned at 20-kb resolution reveals the presence of a 1 Mb sized contact domain with anchors overlapping the *Notch* B2 boundary and a ~100-kb gene dessert downstream of *Notch* both located within chromatin compartment B (Fig. 5a, b). We termed this domain as $N^{enh}$-mega-domain and it shows several prominent features. First, Virtual-4C analysis using as a viewpoint the enhancer at the B2 boundary reveals significant interactions between the B2 boundary and sequences inside and flanking the gene desert (*p*-value < 0.05; Fig. 5b). Second, the B2 boundary contact the full $N^{enh}$-mega-domain in contrast to the boundary at the gene desert (see arrows Fig. 5a). Third, although we observe multiple TADs at $N^{enh}$-mega-domain, we observe physical interactions between TADs within the mega-domain (Fig. 5a) and fourth, the $N^{enh}$-mega-domain is cell type specific (Supplementary Fig. 8a) as the

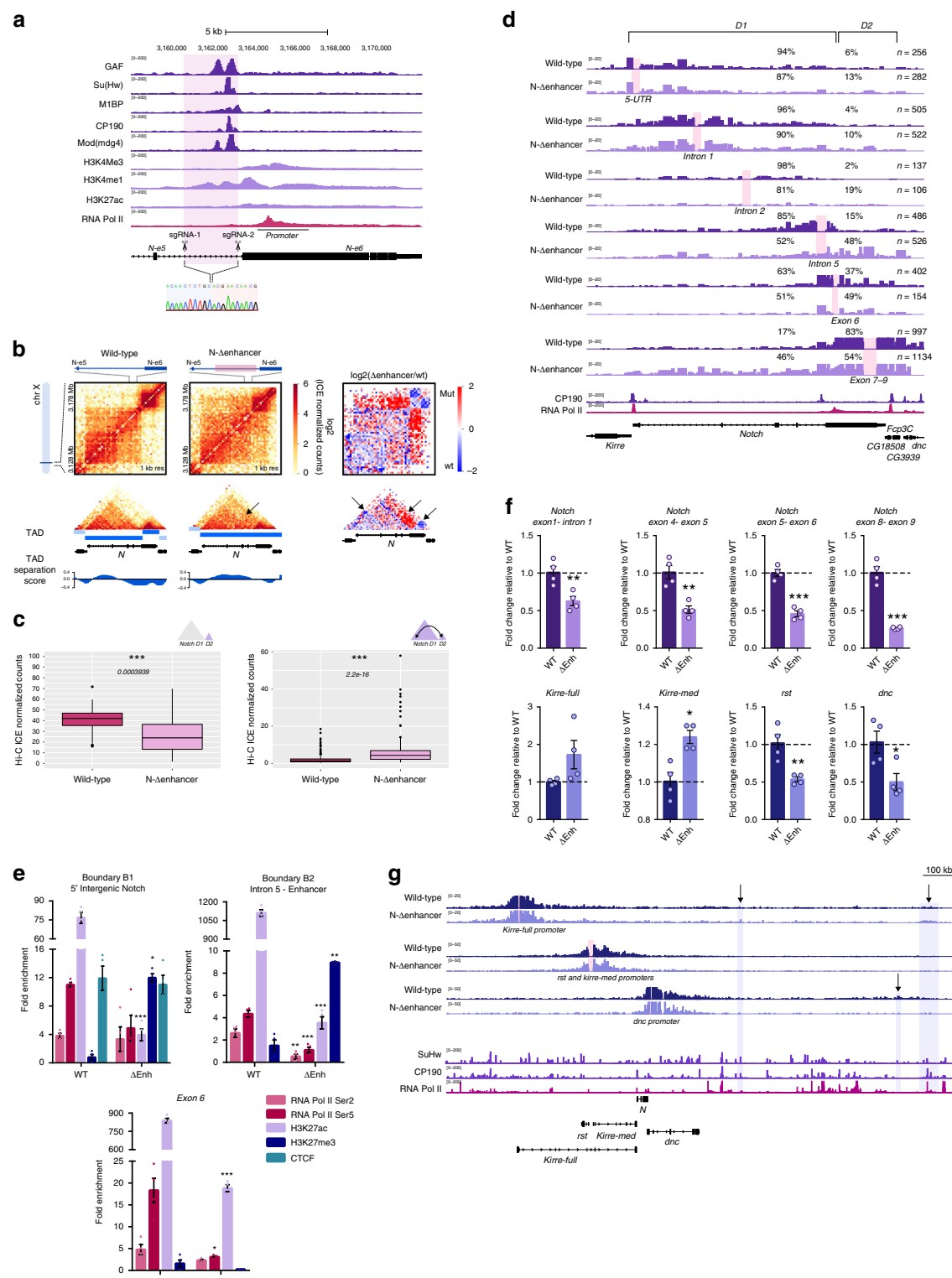

mega-domain boundaries overlap genomic regions with differential chromatin accessibility, enrichment of histone post-translational modifications and binding of APs between cell types (Supplementary Fig. 8b–d).

Deletion of the intronic enhancer of *Notch* results in complete loss of the $N^{enh}$-mega-domain (Fig. 5a). We observed the loss of interaction between the enhancer and the gene desert (HiCcompare $q$-value < 0.01, Fig. 5a) as well as a general loss of intra-domain contacts (Fig. 5a). Interestingly, despite significant

topological effects upon enhancer removal, TADs at the $N^{enh}$-mega-domain and compartments remain mostly invariant upon enhancer deletion (Fig. 5a). Importantly, loss of the $N^{enh}$-mega-domain is specific due to deletion of the intronic enhancer of *Notch* as the mega-domain is unaffected in CRISPR mutants for the 5′ intergenic region of *Notch* (Supplementary Fig. 9a).

Next, we investigated the effect of enhancer deletion and the loss of the $N^{enh}$-mega-domain in gene expression by RNA-seq. We found that 67% of the genes within the mega-domain (34 out

**Fig. 4 Deletion of the intronic enhancer leads to fusion of *Notch* domains and defects in gene expression. a** ChIP-seq tracks for APs, histone marks and RNA Pol II for S2/S2R+ for the intronic enhancer overlapping the B2 boundary. Highlighted is the region deleted by CRISPR-Cas9. An Electropherogram depicting the breakpoint of the deletion is shown underneath. **b** Hi-C normalized heatmaps for wild-type (left) and the enhancer mutant (center) at 1-kb resolution covering a 50-kb region centered in *Notch*. A Hi-C heatmap of the log2 differences in interaction frequency between wild-type and mutant is also shown (right). Arrows indicate regions with changes in interactions. **c** *Notch* D2 intra-domain (left) and *Notch* D1-D2 inter-domain (right) Hi-C counts between wild-type and the enhancer mutant. Significant differences in Hi-C counts was determined by a Wilcoxon rank sum test p-value *$p < 0.05$, **$p < 0.01$, ***$p < 0.001$. Source data are provided as Source Data File. **d** Virtual-4C for wild-type and the enhancer mutant using the 5′ UTR, the Intron 1, Intron 2, Intron 5, exon 6, and the exons 7–9 of *Notch* as viewpoints. Percentages in each track indicate the proportion of interactions between the viewpoint and the *Notch* D1 and *Notch* D2 domains. n, number of valid-pairs for each viewpoint. ChIP-seq tracks for CP190 and RNA Pol II are shown. **e** qChIP experiments in S2R+ wild-type cells and the enhancer mutant. Fold enrichment was calculated against IgG. Significant differences between wild-type and CRISPR mutants were calculated using a t-test. p-value *$p < 0.05$, **$p < 0.01$, ***$p < 0.001$. Error bars represent the standard error of the mean (s.e.m.) of four replicates ($n = 4$). Source data are provided as Source Data File. **f** Top, transcription of *Notch* as measured by RT-qPCR in wild-type and the enhancer mutant using primer pairs spanning the *Notch* D1 domain and one pair at the *Notch* D2 domain. Bottom, Transcription of genes located at TADs flanking *Notch*. To estimate significant differences between samples, a t-test was used. $n = 3$ p-value *$p < 0.05$, **$p < 0.01$, ***$p < 0.001$. Error bars represent the standard error of the mean (s.e.m.) of four replicates ($n = 4$). Source data are provided as Source Data File. **g** Virtual-4C for wild-type and the enhancer mutant using the promoter of the *kirre-full* isoform (top), the promoter of the *rst* and *kirre-med* isoform (middle), and the promoter of *dnc* (bottom) as viewpoints. Arrows indicate regions with ectopic interactions. ChIP-seq tracks for SuHw, CP190, and RNA Pol II are shown.

of 59) are differentially expressed between wild-type and the enhancer mutant (*q*-value < 0.05) (Fig. 5c, d; Supplementary Fig. 9b). Interestingly, the vast majority (31 out of 34) became downregulated by an average of 50% in the enhancer mutant (Fig. 5c; Supplementary Fig. 9b), suggesting that the intronic enhancer could stimulate transcription of genes at the $N^{enh}$-mega-domain, possibly by direct physical interaction as observed in wild-type cells (see arrows, Fig. 5a; Supplementary Fig. 9c) or by diluting regulatory contacts inside the mega-domain upon domain disruption (Fig. 5a). We found among the down-regulated genes, in addition to *Notch*, genes coding for transcription factors like *Myc* and *Mnt*, as well as *tlk* which codes for a serine/threonine kinase that interacts with chromatin regulators and *VhaAC39-1* which promotes *Notch* signaling in imaginal disks. We also validated a subset of differentially expressed genes by qPCR and observed a good agreement with RNA-seq results (Supplementary Fig. 9d), overall supporting an effect on gene regulation for the intronic enhancer of *Notch*.

Taken together our data suggests that the intronic enhancer of *Notch* is a genetic element with two topological activities: it actively insulates the *D1* and *D2* domains of *Notch* while at the same time is responsible for the formation of the $N^{enh}$-mega-domain. Remarkably, loss of the $N^{enh}$-mega-domain impairs gene expression of genes located inside the domain, strongly suggesting that genome topology can impact gene expression.

## Discussion

Here we analyzed the topological and transcriptional consequences of TAD boundary disruption at the *Notch* gene locus in *Drosophila*. We provide evidence that discrete genetic sequences occupied by APs and RNA Pol II are potent chromatin insulators that actively partition the genome into Topological Domains. Furthermore, partial disruption or complete removal of the domain boundaries alter genome topology, transcription, and the recruitment of APs and RNA Pol II. These findings have implications in our understanding of the mechanisms that promote genome organization and the control of gene expression as discussed below.

Whether domain boundaries are autonomous discrete genetic elements mediating the formation of TADs is a subject of intense debate[2]. Our collection of CRISPR-Cas9 mediated deletions of domain boundaries at the *Notch* locus provide evidence on the existence of autonomous genetic elements bound by APs and Pol II that act as chromatin insulators essential for TAD formation.

We found that a 300-bp sequence comprising the entire intergenic region between *kirre* and *Notch* is a modular

chromatin insulator constituting a domain boundary. We uncovered that non-overlapping portions of the intergenic region, with binding sites for specific APs and RNA Pol II, act as discrete modules that restrain interactions of the *kirre* and the *Notch* genes, with the removal of all modules necessary for TAD fusion (Fig. 6). The topological effects observed upon boundary deletion are remarkably consistent with cytological data from the *Notch* mutant *facet-strawberry* (*fa^{swb}*) where deletion of a ~0.9-kb region spanning the 5′ region of *Notch* results in loss of an interband and fusion of the 3C7 band containing *Notch* with the upstream band[26]. Also, reporter assays in transgenic flies and cytological evidence support an autonomous role for the 5′ intergenic region of *Notch* as a chromatin insulator as the ectopic insertion of this sequence is sufficient and necessary to split a band into two, forming an interband in polytene chromosomes[45,46]. In the case of the intragenic enhancer boundary, deletion of a ~2-kb region results in a dramatic increase of ectopic interactions between *Notch* domains and loss of a ~1-Mb domain downstream of *Notch* (Fig. 6). Therefore, evidence from cytological studies in the fly and our *in nucleus* Hi–C data from CRISPR mutants conclusively demonstrates that domain boundaries are essential for TAD formation.

Recent reports have suggested a prominent role for transcription as the main driver for domain organization in *Drosophila*[20,22]. Also, a role for RNA Pol II in mediating domain formation has been recently proposed[47]. Our data provide important observations that support a role for RNA Pol II in boundary activity and therefore in TAD formation in *Drosophila*. First, re-analysis of public Hi–C data from early stages of *Drosophila* embryogenesis suggests that the 5′ boundary of *Notch* is established before Zygotic Genome Activation (nuclear cycle 13), and therefore, before transcription at the locus (Supplementary Fig. 2a). The appearance of TAD boundaries at *Notch* strongly correlates with the early acquisition of chromatin accessibility (nuclear cycle 11) and with the binding of proteins like RNA Pol II (nuclear cycle 12), the general transcription factor TBP, and the pioneering factor Zelda (Supplementary Fig. 2b). Second, transcription inhibition early in development results in a decrease in intra-domain interactions within the *Notch* locus (Supplementary Fig. 2c, d). However, the boundaries and the domains at *Notch* are still detected, which correlates with the retention of RNA Pol II at domain boundaries (Supplementary Fig. 2c, d) suggesting that RNA Pol II is key for TAD formation[47]. In support of this, we observed that deletion of the Promoter Proximal Region of *Notch* (*5pN-Δ102*) resulted in a major decrease in transcription within the *D1* domain (>80%) but just in discrete topological changes

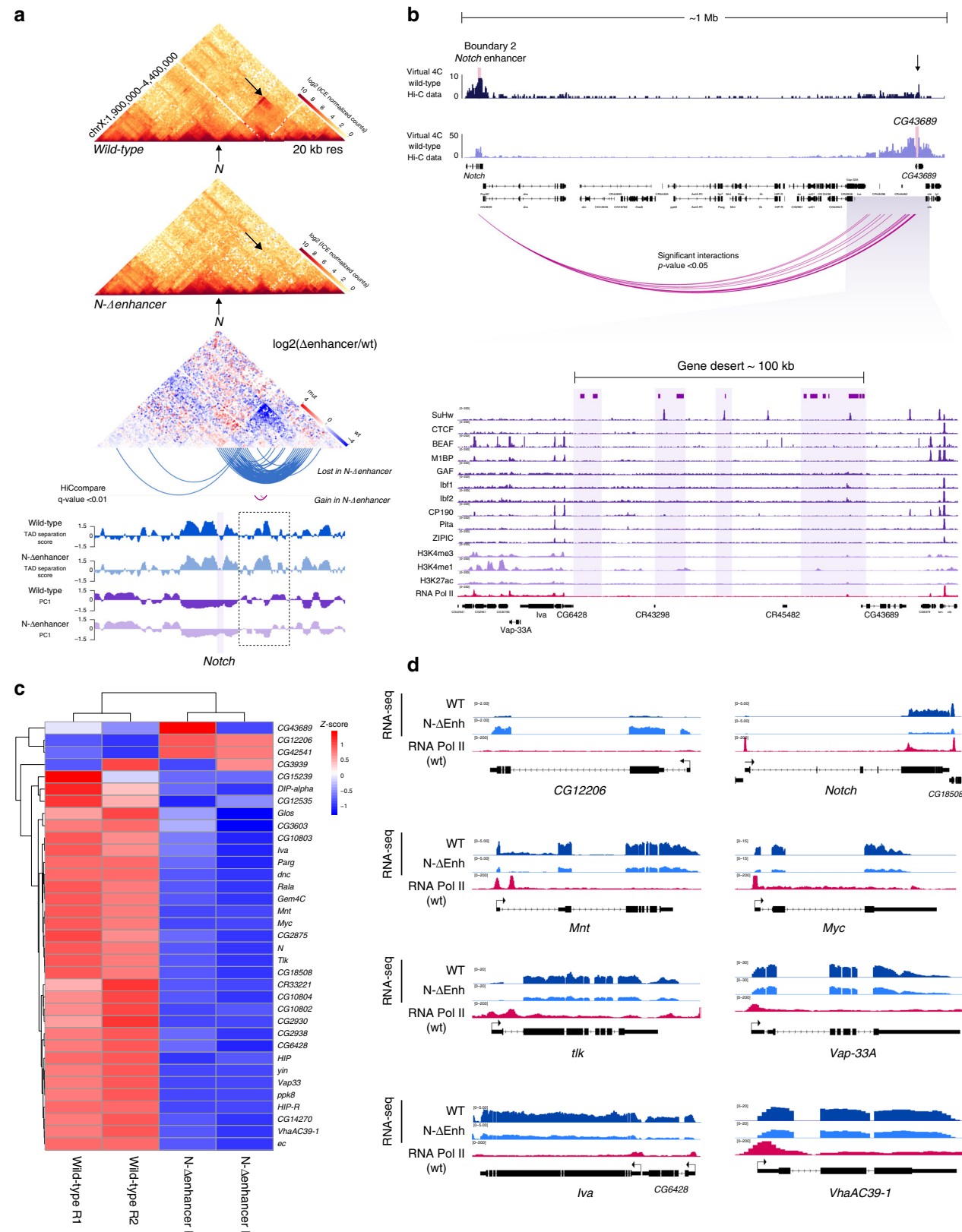

mainly detected as a reduction in intra-domain interactions for the *D1* domain (Fig. 2a; Supplementary Fig. 6a, e), consistent with the topological effects observed at the locus upon transcription inhibition. Importantly, loss of the Promoter Proximal Region of *Notch* resulted in the reduction but not loss of RNA Pol II

binding, which implies that the remaining RNA Pol II could be sufficient to sustain boundary activity. In support of this, the fusion of the *D1* domain of *Notch* with the upstream TAD correlates with complete loss of RNA Pol II at the 5′ end of *Notch*. We observe a similar trend when removing the intronic boundary

**Fig. 5 Deletion of the intronic enhancer of *Notch* disrupts a megabase-sized domain and affects gene expression. a** Hi-C heatmaps at 20kb resolution for wild-type and the enhancer mutant. A Hi-C heatmap of the log2 subtraction of Hi-C matrices between wild-type and the mutant is displayed underneath. HiCcompare detected loss and gain of interactions (arcs, *q*-value < 0.01). Tracks for TAD separation score and PC1 are shown. Dotted box highlight a region with changes in PC1 between wild-type and mutant. **b** *Top*, Virtual-4C for wild-type using the B2 boundary of *Notch* (Enhancer-Intron 5) and the promoter of the *CG43689* locus as viewpoints. *Middle*, Regions of significant interaction with the intronic enhancer of *Notch* in wild-type Hi-C data detected at the fragment level using Seqmonk (*p*-value < 0.01). *Bottom*, ChIP-seq tracks for APs, histone marks and RNA Pol II. Highlighted regions correspond to fragments interacting with the intronic enhancer. **c** Heatmap of the differentially expressed genes inside the mega-domain in wild-type and the enhancer mutant. Expression data were z-score normalized. **d** RNA-seq signal for wild-type and mutant cells for a subset of differentially expressed genes at the mega-domain. ChIP-seq tracks for RNA Pol II are shown.

of *Notch*, with the fusion of *Notch* domains strongly correlating with loss of RNA Pol II binding at the region adjacent to the deleted boundary and in exon 6. Furthermore, we observe the formation of a new TAD spanning the full *Notch* locus despite a significant loss of transcription along the gene (Fig. 4b, f; Fig. 5d). Therefore, although transcription plays a role in mediating intra-domain interactions, our data suggest that discrete, accessible genomic sequences occupied by RNA Pol II, could have a major role in shaping *Drosophila* genome organization independent of transcription.

Architectural Proteins in *Drosophila* can mediate long-range interactions however their role in shaping TADs has remained elusive[48–51]. Our data suggest a role of APBSs in boundary activity in part throughout RNA Pol II recruitment. For example, non-overlapping regions of the 5′ boundary have a differential effect on RNA Pol II recruitment, which correlates with the presence of different APBSs (Supplementary Fig. 4a). In particular, we observed that deletion of a ~200-bp region containing just a CTCF motif (*5pN-Δ183*) have a stronger effect in RNA Pol II recruitment than deletion of the Promoter Proximal Region (*5pN-Δ102*) which contains a binding site for M1BP (Fig. 2f). Then, in this case the CTCF DNA-binding motif seems important to either directly or indirectly recruit RNA Pol II. In support of this, mutation of the CTCF motif results in loss of binding of nuclear proteins (Supplementary Fig. 6g). We also observed that loss of both CTCF and M1BP binding sites in the *5pN-Δ343* mutant correlates with the maximal decrease of CTCF and RNA Pol II occupancy, complete loss of insulation and TAD fusion, implying that domain boundaries can be resilient to the loss of RNA Pol II binding through the presence of multiple APBSs. In support of a role for DNA-binding APs in boundary activity through RNA Pol II recruitment, Hug et al. reported that depletion of the pioneering factor Zelda results in loss of RNA Pol II recruitment, deficient local insulation and fusion of adjacent TADs[23].

TAD boundaries can block unspecific regulatory communication[10], however, their role in gene regulation has been recently subject to intense debate[12,52]. Our data support that TADs have an important function in gene regulation in *Drosophila*. We found that deletions spanning the 5′ boundary of *Notch*, consistently results in loss of transcription within the *D1* domain likely as a combination of reduced RNA Pol II occupancy at the 5′ end of the gene, loss of insulation between adjacent TADs and gain of ectopic interactions (Fig. 6). Deletion of the B2 boundary also results in a reduction in *Notch* transcription. Interestingly, boundary disruption leads to loss of RNA Pol II binding at exon 6 and at the 5′ region of *Notch*, suggesting that it influences RNA Pol II recruitment to the *Notch*, locus probably by direct physical interaction. Then, in this case, reduction in transcription could be a consequence of disrupting physical interactions between regulatory elements that affect RNA Pol II recruitment, rather than the consequence of insulation loss. Furthermore, loss of the mega-domain due to deletion of the B2 boundary affects gene regulation of the genes located within the domain. Therefore, our

evidence show that disruption of TAD organization by alteration of boundaries impacts gene expression.

Finally, an important observation from our experiments is that deletion of TAD boundaries and accompanying changes in gene transcription as well as changes in the recruitment of CTCF and RNA Pol II at domain boundaries, do not abolish the intra-TAD specific organization of *Notch* since subdomains are preserved despite TAD fusion (See Fig. 2a and Fig. 4b). These suggest that additional mechanisms contribute to folding the genome into smaller domains, possibly by aggregation of regions with similar chromatin features as has been suggested[4,20,22].

In conclusion, our data demonstrate the existence of discrete genetic sequences with boundary activity that influence genome organization into Topological Associated Domains and the regulation of gene expression. Other domain boundaries with a similar chromatin composition and APs occupancy could behave similarly.

Based on our results, we propose a mechanism for boundary formation through the binding of APs that results in recruitment of RNA Pol II. In such a model, a boundary is robust to APs depletion as far as RNA Pol II binding is maintained. Finally, we envision that genome organization in *Drosophila* is dependent on two mechanisms: one driven by self-association of regions with similar transcriptional or epigenetic profiles and one that partitions the genome into interaction domains driven by genetic elements acting as chromatin insulators.

## Methods

**Cell culture**. *Drosophila* S2R+ cells (Drosophila Genomics Resource Center (DGRC)) were cultured at 25 °C in Schneider's *Drosophila* Medium (Thermofisher) supplemented with 10% FBS (Biowest) and penicillin/streptomycin (Invitrogen).

**Antibodies**. The following antibodies were used for ChIP: dCTCF polyclonal antibodies were generated by New England Peptide by immunizing rabbits with a peptide corresponding to the first 20 aminoacids of dCTCF[53]. The following antibodies were obtained from commercial sources: anti-H3K4me3 (Abcam #8580), anti-H3K27ac (Abcam #4729), anti-H3K27me3 (Abcam #6002), anti-RNA Pol II pSer2 (Abcam #5095), anti-RNA Pol II pSer5 (Abcam #5408), anti-IgG mouse (Millipore #12-371), and anti-IgG Rabbit (Millipore #12-370).

**CRISPR-Cas9**. *Design and cloning of sgRNAs into pAc-sgRNA-Cas9*. For design and evaluation of guide sequences for the CRISPR-Cas9 system we used the CRISPROR software (http://crispor.tefor.net/)[54] and the *Drosophila melanogaster* dm6 version of the genome. The genomic regions used for guide design were chrX:3,134,000-3,134,869 and chrX:3,160,210-3,163,651 which correspond to the 5′ region of *Notch* and the intronic enhancer, respectively. Selected guide sequences had among the lowest off-target scores. Sequences for all guides used in this study are provided in Supplementary Table 2. Importantly, the guides used to generate deletions over the 5′ region of *Notch* do not target the sequence containing the major TSS sites annotated for *Notch* and the guides for the intronic enhancer are located at least 100 bp away from the nearest exon. Oligonucleotides for guides were cloned into pAc-sgRNA-Cas9 (Addgene #49330) as described[55]. Integrity of cloned guides was evaluated by Sanger Sequencing. Plasmids were expanded and purified using a Qiagen MiniPrep kit prior to transfection.

*Transfections*. CRISPR-Cas9 mediated deletions were generated by transfecting two plasmids each with a specific guide targeting each side of the region of interest (See Fig. 3). Transfection was carried out in S2R+ cells as described with major modifications[55]. Briefly, S2R+ cells were seeded at 600,000 cells/well in a 24-well plate in 500 μl of Schneider's *Drosophila* Medium before transfection. FugeneHD

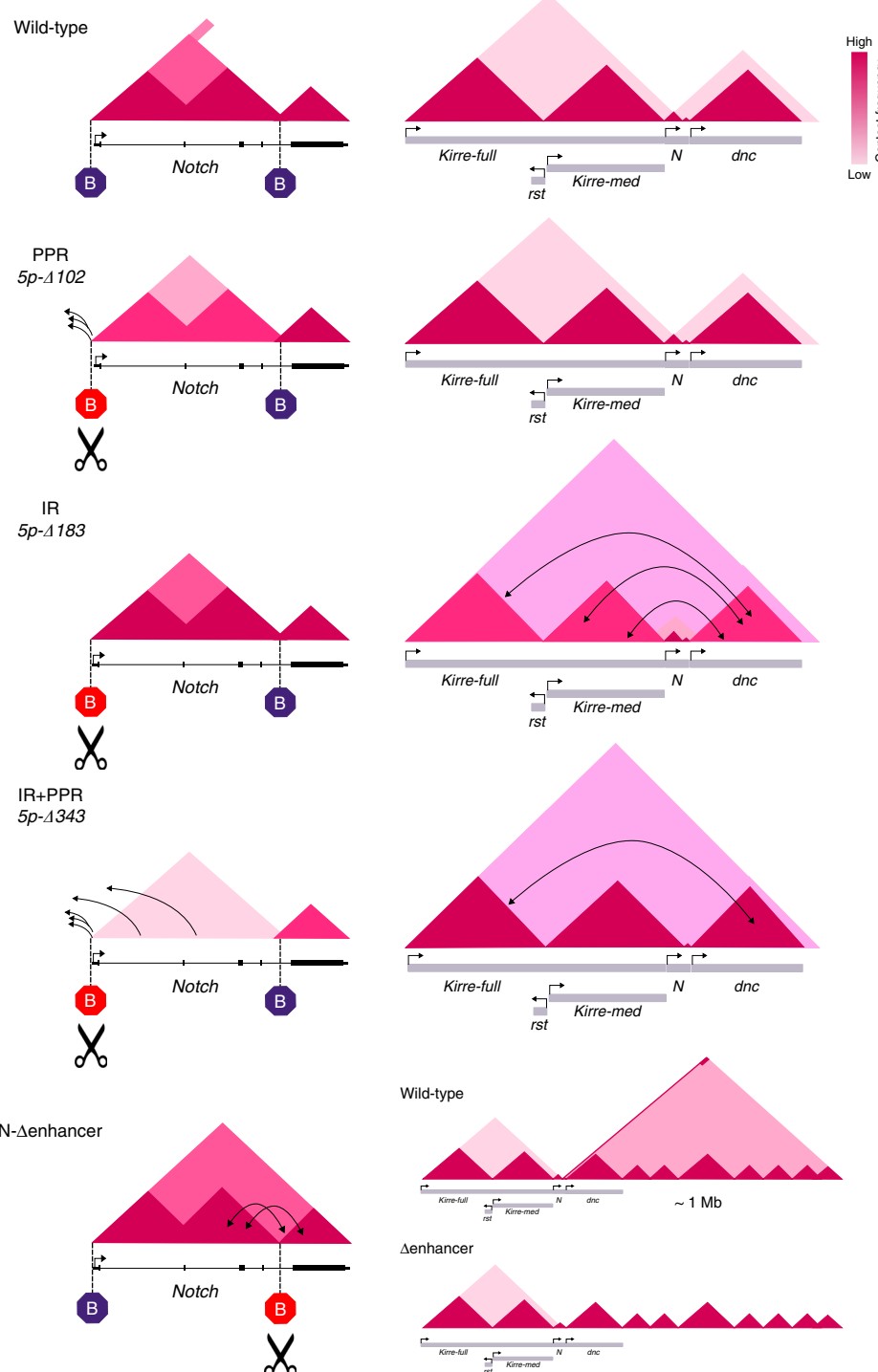

**Fig. 6 Domain boundaries actively partition the *Notch* gene locus into TADs and mediate the formation of a megabase-sized domain.** The *Notch* gene locus is organized into two TADs with boundaries overlapping the 5′ intergenic region of *Notch* and an intronic enhancer. Deletion of the Promoter Proximal Region (PPR) results in loss of intra-domain interactions at the D1 domain of Notch and is accompanied by ectopic interactions of the 5′ UTR with the upstream TAD. Deletion of the Intergenic Region just upstream of the PPR (IR) leads to a major gain of ectopic interactions between the TADs flanking *Notch*. Deletion of the full intergenic region of Notch (IR + PPR) results in TAD fusion and gain of ectopic interactions between TADs flanking *Notch*. Deletion of the intragenic enhancer results in fusion of *Notch* domains and loss of a megabase-sized domain downstream of *Notch*.

(Promega) was used for transfection using manufacturer guidelines (https://worldwide.promega.com/techserv/tools/FugeneHdTool/). A 4:2 FugeneHD:Plasmid ratio was used for all transfections. Transfected cells were placed at 25 °C and 72 h after transfection cells were re-suspended in fresh medium supplemented with 5 μg/mL of Puromycin (Sigma) and transferred into a well of a 12-well plate. After 3 days (6 days post-transfection), cells were re-suspended in fresh medium supplemented with 5 μg/mL of Puromycin (Sigma) and transferred into a well of a

6-well plate. After 3 days (9 days post-transfection) an aliquot of cells was used for DNA extraction by phenol-chloroform and PCR genotypification using specific primers spanning the desired deletions (see Supplementary Table 2). Pools of mutant cells were expanded in fresh Schneider's *Drosophila* Medium without Puromycin and used for serial dilution and isolation of clones of mutant cells.

*Clonal isolation of mutant cells.* Pools of mutant cells were used for serial dilution as described in with minor modifications[56]. Mutant cells were re-

suspended in Schneider's *Drosophila* Medium conditioned medium (30% medium from mutant cells growth for 72 h and 70% of fresh Schneider's medium with 10% FBS and penicillin/streptomycin) and 100 μl of medium with cells were added per well of 96-well plate. Cells were incubated at 25 °C for at least 3 weeks. Single clones were expanded and an aliquot of cells was used for DNA extraction by phenol-chloroform and PCR genotypification using specific primers spanning the desired deletions (Supplementary Table 2). Homozygous clones were expanded and used for subsequent experiments. Mutant clones were further characterized by Sanger Sequencing using primers for genotypification. Amplified fragments were ligated into pGEM-T Easy and two individual clones were used for Sanger Sequencing per mutant. The sequence of each mutant fragment was the same between clones and Electropherograms of breakpoints for all CRISPR mutants generated in this study are shown in Supplementary Fig. 4. Importantly, since S2R+ cells are tetraploid, since each cut by the CRISPR-Cas9 system is independent at each allele we cannot rule out the existence of specific indels resulting at each deletion junction at each allele, that can be detected either by visual inspection of agarose gels nor in the sequenced clones by Sanger Sequencing.

**In nucleus Hi–C data generation and processing**. *In nucleus Hi–C and sequencing. in nucleus* Hi–C libraries were generated for S2R+ wild-type cells and CRISPR mutants at least in duplicates using an *in nucleus* Hi–C protocol with minor modifications for S2R+ cells[29]. Briefly, $40 \times 10^6$ S2R+ cells were fixed with formaldehyde at a final concentration of 2% for 10 min at room temperature and fixing reaction was stopped by adding 1 M Glycine for 5 min rocking. Cells were washed with cold-PBS and flash frozen for short-term storage at −70 °C or used immediately for subsequent experiments. The genomes were digested with MboI (NEB) and 5′ overhangs were filled with Biotin-14-dATP (Invitrogen). Hi–C libraries were sequenced on a HiSeq 2500 platform 50 pb paired-end.

*Generation of mutant genomes in silico.* Individual in silico mutant genomes were built for all CRISPR mutants to allow for precise mapping of reads without gaps. The sequence of breakpoints generated at deletion sites for each CRISPR mutant (as determined by Sanger Sequencing) were used to replace the wild-type sequence for the resulting mutant sequence in the chromosome X using R. A specific index for each mutant genome was build using Bowtie2 (ref. [57]). Also, mutant genomes were used to generate mutant-specific MboI digestion files for subsequent analysis using HiC-Pro digest_genome.py.

*Data processing.* Mapping, filtering, correction and generation of Hi–C matrices were done using HiC-Pro[58]. Briefly, read pairs were mapped independently to the wild-type or mutant genomes generated in silico using Bowtie2 with HiC-Pro default parameters. After filtering, valid-pairs were used to generate raw and ICE normalized matrices at 1 kb, 5 kb, 20 kb, and 100 kb bin resolution. To correct for differences in sequencing depth matrices were normalized by the smallest number of valid-read pairs (Supplementary Table 1). All heatmaps of normalized contact matrices were generated with HiCPlotter[59]. Correlation plots between all Hi–C replicates and counts vs distance plots were generated using HiCExplorer[19] and matrices with a bin size of 10 kb.

*TAD calling.* TADs were identified for matrices at 1 kb and 5 kb resolution for wild-type and all CRISPR mutants using the TAD separation score from HiCExplorer. Identification of TADs at 1 kb resolution was done with parameters –minDepth 10000 –maxDepth 40000 –step 1500 –thresholdComparisons 0.0000001 –delta 0.01 and –correctForMultupleTesting fdr. Identification of TADs at 5 kb resolution was done with parameters –minDepth 20000 –maxDepth 200000 –step 10000 –thresholdComparisons 0.01 –delta 0.01 and –correctForMultupleTesting fdr. TAD separation score bedgraph files were displayed using IGV[60]. Analysis of TAD conservation between samples was done using Intervene[61] with options venn –bedtools-options $f = 0.8$. Boxplots of Hi–C counts for intra and inter-domain interactions were generated in R using normalized matrices from HiC-Pro and TAD coordinates for the wild-type sample obtained from HiCExplorer. A Wilcoxon-Rank Sum Test was used to determine statistically significant differences.

*Virtual 4C analysis.* Virtual4C profiles for different viewpoints were generated by using HiC-Pro *make_viewpoints.py* and the same number of valid-pairs for the datasets of interest. IGV was used to display bedgraph files. The percentage of valid-pairs with a viewpoint of interest at a specific genomic region was calculated using the R software (http://www.R-project.org).

*Compartment analysis.* Compartments were identified using HOMER[62] *runHiCpca.pl* with parameters –res 10000 –window 50000. Peaks of enriched for the histone mark H3K4me3 in S2R+ cells (GSM2259985) were identified using MACS2 (ref. [63]) and used to assign sign to the PCA1 results. PCA1 score bedgraph files were displayed using IGV[60]

*Differential analysis of Hi–C datasets.* HiCcompare[64] was used to identify differences in chromatin interactions with default parameters. Pairwise comparisons between wild-type and CRISPR mutants was performed using raw contact matrices at 5 and 20 kb resolution as input. Differential contacts in regions of interest were plotted using the Washington Epigenome Browser.

*Hi–C data processing of public datasets.* Publicly available Hi–C datasets from *Drosophila* embryos were obtained from the European Nucleotide Archive (ENA) database under the following accession numbers: embryos nuclear cycle 12 (ERR1533189-199), embryos nuclear cycle 13 (ERR1533200-209), embryos nuclear cycle 14 (ERR1533226-236), embryos 3–4 h (ERR1533170-181), and embryos

nuclear cycle 14 triptolide treatment (ERR1912894-899). Publicly available Hi–C datasets from *Drosophila* cell lines were obtained from the Gene Expression Omnibus (GEO) database under the following accession numbers: Kc167 cells (GSM2133771, GSM1551441, GSM1551442, GSM1551443, and GSM1551444) BG3 cells (GSM3475690 and GSM3475691). Datasets were processed as described using Bowtie 2 and HiC-Pro. Heatmaps from contact matrices were generated from HiCPlotter.

**RNA-seq data generation and processing**. Total RNA was extracted from wild-type and enhancer mutants in duplicate using TRIzol reagent (Thermofisher) following manufacturer's instructions. Total RNA libraries for all samples were generated and sequenced by Novogene. Two independent libraries per condition were sequenced in a HiSeq 4000 platform paired-end 150 bp. RNA-seq data was analyzed using Salmon[65] and DESeq2 (ref. [66]). Briefly, we estimated transcript-level abundance for each dataset using Salmon with a specific index for the *Drosophila melanogaster* transcriptome (dm6) and options –validateMappings and --gcBias. Quantification data from Salmon was then imported into R and we created an input table with gene level quantification data as input for DESeq2. We tested for differential expression (DE) between wild-type and the CRISPR enhancer mutant using DESeq2. We called DE at the gene level with a fold-change of at least 0.5 and a corrected *p*-value of <0.05. Abundance levels for genes located at the $N^{enh}$-*mega-domain* were Z-score transformed and used to create a heatmap with pheatmap using R. To create signal tracks from RNA-seq data, sequencing reads were mapped against the *Drosophila melanogaster* genome (dm6) using Bowtie2 with default parameters. SAM files were converted into BAM files using samtools view. BAM files were sorted (samtools sort) and indexed (samtools index). deepTools[67] v3.3.0 was used to calculate Pearson Correlation and PCA for all datasets. BAM files from replicates were merged due to high correlation (Pearson Correlation > 0.9) and BAM files from merged experiments were used to create signal tracks with bam-Coverage –normalizeUsing BPM. Signal tracks for wild-type and the CRISPR enhancer mutant data were visualized using IGV.

**RT-PCR**. Total RNA was extracted from wild-type and CRISPR mutants at least in duplicate using Trizol and following manufacturer's instructions. RNA was used directly for qRT-PCR using the KAPA SYBR FAST One Step kit (KAPA Byosystems) in at least two technical replicates per sample using a StepOne Real-Time PCR System. A list of all primer sets used in this study are provided in Supplementary Table 2. Primers were designed to span exon-exon junctions when possible. The constitutively expressed gene *RP49* was used as a control. Data was analyzed by the *ΔΔCt* method[68]. Statistically significant differences in gene expression between the wild-type and the CRISPR mutants were computed using a *T*-test (*p*-value < 0.05) and the Graphpad Prisma Software 7.0.

**ChIP and qPCR**. Chromatin immunoprecipitation (ChIP) using wild-type cells and all CRISPR mutants was performed in duplicate as previously described[69]. For each IP, the following amount of antibody was used: 5 μg/IP for anti-CTCF, anti-IgG rabbit and anti-H3K27me3; 2 μg/IP for anti-H3K4me3, anti-H3K27ac, anti-RNA Pol II pSer5, pSer2 and anti-IgG mouse. Purified DNA was used for region-specific quantification by qPCR using SYBR Green in duplicate per ChIP. Mean values for all the regions analyzed in different conditions were expressed as fold-enrichment compared over IgG. Statistically significant differences between wild-type samples and CRISPR mutants were computed using a *T*-test (*p*-value < 0.05) and the Graphpad Prisma Software 7.0. ChIP primers are listed in Supplementary Table 2.

**EMSA**. Electrophoretic Mobilitiy Shift Assay (EMSA) was performed as previously described with minor modifications[70]. In particular, nuclear protein was extracted using ~$40 \times 10^6$ *Drosophila* S2R+ cells. For super-shift assays 2 μg of dCTCF or IgG rabbit antibody were used. Oligo-sequences used for EMSA assays are listed in Supplementary Table 2.

**ChIP-seq and ATAC-seq data retrieval and processing**. Publicly available ChIP-seq data for S2/S2R+ cells and *Drosophila* embryos were obtained from the Gene Expression Omnibus (GEO) database. Raw data for S2/S2R+ cells was obtained under the following accession numbers: CP190 (GSM1015404), SuHw (GSM1015406), Mod (GSM1015408), CTCF (GSM1015410), Ibf1 (GSM1133264), Ibf2 (GSM1133265), BEAF32 (GSM1278639), Pita (GSM1313420), ZIPIC (GSM1313421), RNA Pol II (GSM2259975), H3K4me1 (GSM2259983), H3K4me3 (GSM2259985), H3K27ac (GSM2259987), MSL2 (GSM2469507), H4K16ac (GSM2469508), M1BP (GSM2706055), GAF (GSM2860390), Input (GSM1015412). Raw data for Kc167 cells was obtained under the following accession numbers: CTCF (GSM762842), BEAF32 (GSM762845), Ibf1 (GSM2133766), Ibf2 (GSM2133767), Pita (GSM2133768), ZIPIC (GSM2133769), Su(Hw) (GSM762839), GAF (GSM2133762), CP190 (GSM762836), RNA Pol II (GSM1536014), and H3k27ac (GSM890121). ATAC-seq data from *Drosophila* cell lines was obtained from GEO under the following accession numbers: Kc167 (GSM3381113), S2 (GSM3381126).

Raw data for *Drosophila* embryos was obtained under the following accession numbers: Zelda nc8 (GSM763060), Zelda nc13 (GSM763061), Zelda nc14 (GSM763062), TBP MBT (GSM1022898, GSM1022899), TBP post-MBT

(GSM1022903, GSM1022911), TBP pre-MBT (GSM1022912), RNA Pol II-pSer5-nc12 (GSM1536376), RNA Pol II-pSer5-nc13 (GSM1536379), RNA Pol II-pSer5-nc14early (GSM1536382), RNA Pol II-pSer5-nc14mid (GSM1536384), RNA Pol II-pSer5-nc14late (GSM1536386), and RNA Pol II-pSer5-nc13ZeldaKD (GSM1536390). Raw data from *Drosophila* embryos was obtained from ENA under the following accession numbers: RNA Pol II-pan nc14 (ERR1912880 and ERR1912881), RNA Pol II-pSer5 nc14 (ERR1912882 and ERR1912883), RNA Pol II-pan nc14-triptolide (ERR1912890 and ERR1912891), and RNA Pol II-pSer5 nc14-triptolide (ERR1912892 and ERR1912893). ATAC-seq data from *Drosophila* embryos was obtained from GEO under the following accession numbers: ATAC-nc11-3 h(GSM2219678), ATAC-nc11-6 h(GSM2219681), ATAC-nc11-9 h (GSM2219684), ATAC-nc12-3 h(GSM2219687), ATAC- nc12-6 h(GSM2219690), ATAC- nc12-9 h(GSM2219693), ATAC- nc12-12 h(GSM2219696), ATAC- nc13-3 h(GSM2219700), ATAC- nc13-6 h(GSM2219703), ATAC- nc13-9 h (GSM2219706), ATAC- nc13-12 h(GSM2219709), ATAC- nc13-15 h (GSM2219712), and ATAC- nc13-18 h(GSM2219715).

Reads were mapped against the *Drosophila melanogaster* genome (dm6) using Bowtie2 with default parameters for single and paired reads. Mapped reads were filtered by map quality (-q 30) using samtools (sammtools view). Bam files were sorted (samtools sort) and indexed (samtools index). Duplicates were removed with Pickard. Bam files were imported to deeTools v3.3.1 to create signal tracks with bamCoverage –normalizeUsing CPM. Signal tracks for all data were visualized using IGV. Peak calling for Architectural Proteins in S2/S2R+ cells was performed using MACS2 (ref. [63]) *callpeak* function with option –call-summits and default parameters.

**modENCODE data sets**. ChIP-chip signal files (.wig) were retrieved from http://www.modencode.org/ under the following accession numbers: H4K16ac-Kc167 (318), H4K16ac-S2 (319), H4K16ac-BG3 (316); H3K27ac -S2 (296), H3K27ac -BG3 (295); H3K27me3 -Kc167 (5136), H3K27me3-S2 (298), and H3K27me3-BG3 (297). Wig files were visualized using IGV.

**Motif analysis**. Identification of binding sites for insulator proteins was done using FIMO[71] and the JASPAR[72] Insect Database using as a threshold *p*-value < 0.0001.

**Reporting summary**. Further information on research design is available in the Nature Research Reporting Summary linked to this article.

## Data availability

Raw data for *in nucleus* Hi–C and RNA-seq experiments generated in this study have been deposited in the Gene Expression Omnibus (GEO) repository as a SuperSeries under the accession number: GSE136137. Publicly available Hi–C, ChIP-seq, and ATAC-seq datasets were obtained using accession numbers provided in the Methods section. All other relevant data supporting the key findings of this study are available within the article and its Supplementary Information files or from the corresponding authors upon reasonable request. The source data underlying Figs. 1b, c, 2f, 3b, c, and 4c, e, f, Supplementary Figs. 6a–d, f, 7c, d, and 8d are provided as a Source Data File. A reporting summary for this Article is available as a Supplementary Information file.

## Code availability

Custom code used in this manuscript is available at https://github.com/RodrigoArz/In-situ-dissection-domain-boundaries-in-Drosophila--Custom-Code.

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

## Acknowledgements

Work in the authors' laboratories is supported by a grant from Consejo Nacional de Ciencia y Tecnología (Fronteras de la Ciencia 2015-290) and Fundación Miguel Alemán to F.R-T. and by Programa de Apoyo a Proyectos de Investigación e Innovación Tecnológica (DGAPA-PAPIIT IN203917 and IN203620) to F.R.-T. and (DGAPA-PAPIIT IN207319) to M.F-M. We are grateful with the Molecular Biology Unit at the Instituto de Fisiología Celular, UNAM for the services provided for the sequencing experiments. R.G. A.-M. is a doctoral student from Programa de Doctorado en Ciencias Biomédicas, Universidad Nacional Autónoma de México, and received fellowship 288814 from Consejo Nacional de Ciencia y Tecnología (CONACyT). We are grateful to Victor Corces for sharing information about the 3D organization of *Notch* from Hi-C data and for support with establishing the CRISPR-Cas9 system in *Drosophila* cells. R.G.A-M is grateful to Victor Corces for mentoring and support during his Ph.D. as well as to Martha Vazquez Laslop for mentoring and intellectual contributions during the development of this project.

## Author contributions

R.G.A.-M., M.F.-M., and F.R.-T. designed the project. R.G.A.-M. performed all experiments and computational analysis with the support of A.J.C.-C., G.G., and M.F.-M.; R.G. A.-M., M.F.-M., and F.R-T. wrote the manuscript.

## Competing interests

The authors declare no competing interests.
