## [Peer Review File · Nature Communications]

Reviewers' comments:

Reviewer #1 (Remarks to the Author):

The relationship between transcription, architectural proteins and the three-dimensional organization of the genome is a topic of great interest. In this manuscript, the authors address this issue by performing targeted deletions of architectural protein sites and regulatory sequences in the Notch gene of *Drosophila*. Authors find that both architectural proteins and proteins involved in transcription contribute to 3D organization. The results are interesting and appropriate for publication in *Nature Communication*. However, authors should first address the following issues:

1. Figure 1b. Is it known whether the "main" and exonic promoters of Notch are transcribed in the cells used for these analyses? It is interesting that Domain 1 lacks H3K27ac, Pol II, and H3K4me3, while Domain 2 contains all these marks. It would be interesting to show the distribution of H3K27me3, even if it is from a related cell line. The presence of H3K7me3 in domain 1 but not domain 2 would suggest that the two domains are formed by their different transcriptional states.
2. "data suggest that the Promoter Proximal Region of Notch is necessary to promote intra-TAD interactions at D1 domain and to restrict communication between the 5'UTR with the neighboring TAD". Authors should be careful with this type of interpretation. If I cut my hand off, for example, and I start bleeding profusely from the residual arm, this result does not mean that the normal role of the hand is to restrict bleeding from the arm. The role of the promoter-proximal region of Notch is to recruit proteins necessary for transcription.
3. "The increase in inter-domain interactions is accompanied by a decrease in the number of interactions within each domain, likely reflecting the engagement of their sequences in ectopic contacts (Fig. 2d; Supplementary Fig. 5a)". This is certainly possible but, more likely, since the comparison between the two samples requires normalizing for the number of reads, if something goes up something else has to come down, no matter what.
- 4 "Moreover, visual inspection of Hi-C heatmaps at 5 kb resolution suggest that inter-domain interactions are constrained by the 5' and 3' boundaries of the kirre domain-1 and the dnc domain, respectively (Fig. 2d)". Here and throughout the manuscript, authors assume that domains are formed by a boundary that does not allow interactions to occur between the two adjacent domains. This is what happens between two adjacent CTCF loops in mammals, where continuous extrusion by cohesin may interfere with interactions between the two loops. However, it is unclear whether this is also the case in *Drosophila*. More likely, the boundary is formed as a consequence of sequences on either side of the boundary binding to proteins that, because of their biochemistry, prefer to interact with other proteins present on sequences located either to the left or the right of the boundary. Therefore, it is not that there is a barrier to interactions. Rather, proteins actively determine the direction of interactions based on the location of their chemically preferred partners. If, as the authors suggest, architectural proteins form a barrier to interactions, what would be the mechanism by which this happens?

5. Although the sites for architectural proteins appear to be together based on Figure 1A, there are clear differences between the locations of the peak summits. For example, GAF does not seem to overlap with SuHw or CTCF. It may help in the interpretation of the phenotypes if the authors show a figure panel, similar to Figure 1D, showing the deleted sequences and the location of the binding motifs of the different architectural proteins, which should be located at the summits of the peaks. This will complement results in Figure 2F and also allow the authors to interpret the deletions in the context of the different architectural proteins affected. This information may be important in the interpretation of the results. For example, D102 appears to delete the site for M1BP but not CTCF, D183 deletes the site for CTCF but not M1BP, and D343 appears to delete both. If available, ATAC-seq data may allow the mapping of binding sites for other proteins in the region, specially those potentially bound to sequences affected by the D755 deletion.

6. "A similar topological effect is observed with the 5pN- Δ 755 mutant, although less severe (Fig. 2a; Supplementary Fig. 5a)." This deletion affects a larger region of the genome than D343, but the effect is less severe. One possibility would be that sequences located between the right breakpoints of D755 and D343 i.e. sequences containing the binding site for M1BP and potentially other proteins are responsible for the increased effect of D343. As suggested in comment #7, paying attention to the specific binding sites deleted by the various deletions may be important in the interpretation.

7. "The marked decrease in RNA Pol II binding at the B1 boundary in the 5pN- Δ 183 mutant suggest that the 5' intergenic region of Notch is essential for RNA Pol II recruitment". The intergenic region is larger than the region covered by D183. The results only suggest that sequences contained within the D183 deletion are important for Pol II recruitment.

8. Figure S5D. It is unclear how the authors can conclude that the data in this figure indicate that CTCF is specifically required to recruit Pol II.

9. Figure 5A. "Visual inspection of wild-type Hi-C data binned at 20 kb resolution reveals the presence of 1Mb sized Topologically Associated Domain with boundaries overlapping the Notch intronic enhancer and a \sim 100 kb gene desert downstream of Notch (Fig. 5a,b)". I disagree with this interpretation. What the authors see is a compartmental interaction between sequences in Notch and a small domain located on top of the "0" of the "20 kb res" legend of the heatmap. A distance-normalized heatmap or the Pearson correlation will show that the signal corresponds to a compartmental interaction.

10. Figure 5A. "domain boundaries are engaged in a long-range interaction similar to what has been observed for anchors of chromatin loops mediated by Pc and by CTCF". Interactions mediated by CTCF in mammals or Pc in *Drosophila* appear as circular dots of signal with an intense central pixel surrounded by pixels of decreasing intensity. The interaction marked by an arrow in Figure 5a does not look like this. It has the shape of a line, indicating interactions between a short sequence in Notch and a large domain to the right of the map. A rough calculation suggests this domain is

118 kb.

11. "although we observe multiple sub-TADs at Nenh344 -mega-domain". At 1 kb resolution these domains were referred to as TADs. Now, at 20 kb resolution, they are called sub TADs. What determines if a domain is a TAD or a subTAD?

12. "The overall organization of the Nenh346 -mega-domain is similar to loop domains described in mammals, where regions at the loop anchor display a high frequency of interaction while at the same time sequences within the domain interact". As described in comment #10, I disagree with this interpretation. Loop domains in vertebrates are formed by cohesin extrusion and their anchors contain CTCF, which binds to a short sequence of a few bp, rather than a 118 kb domain. This large domain would normally be classified as a compartment.

13. "Together, our data show that the enhancer at the intron five of Notch is a topological insulator that restricts communication between the two topological domains of Notch. Also, it is essential to recruit RNA Pol II at Notch promoters and therefore for Notch transcription". The enhancer of Notch is in an intron and appears to contact both the upstream and downstream promoters. These directional contacts create two small domains on both sides of the enhancer. Interpreting these results to say that the enhancer is "a topological insulator that restricts communication between the two topological domains" is inappropriate. If enhancers are insulators, then we don't need two different words to name these sequences.

14. Figure 5B shows a virtual 4C analysis of interactions between the Notch enhancer and the downstream domain shaded in purple. Are all the genes whose expression is affected by the deletion of the enhancer within this purple region? In Figure 5A it appears that the enhancer contacts many sequences in what the authors call mega domain, not just the ones shaded in purple in Figure 5B. Are these other interactions not statistically significant? Authors should comment on how the enhancer affects the transcription of genes located between the enhancer and the purple shaded domain in Figure 5B i.e. those that appear not be contacted based on the virtual 4C.

Minor comments

1. I'm not familiar with the term "in nucleus Hi-C". Authors should explain how this Hi-C method differs from the commonly used in situ Hi-C.

2. Authors use the term "topological insulator", which I have not seen used in this context before. Is this a new type of insulator, different from those described before? Wikipedia says, "A topological insulator is a material with non-trivial symmetry-protected topological order that behaves as an insulator in its interior but whose surface contains conducting states, meaning that electrons can only move along the surface of the material". Authors should make sure this is what they mean by topological insulator. In general, the authors over-use the term "topological" throughout the manuscript. For example, in these two consecutive sentences "As shown, loss of the B1 boundary does not affect the topological organization of the D2 domain of Notch, which

suggests that it is an independent topological unit. The topological organization of the D2 domain could result from the activity of a topological insulator" the authors use "topological" 4 times. I would just use delete every instance of this word in the manuscript.

3. Figure S5C. It is not clear what the title "CTCF en AbdB" in the figure panel means.

Reviewer #2 (Remarks to the Author):

In this manuscript, Arzate-Mejia et al. dissect the 3D chromatin organisation at Notch locus in *Drosophila* S2 cells. In particular, they show that the locus is organised in two domains separated by a boundary element. They then perform a series of experiments to remove two boundaries and observe the changes in 3D chromatin organisation. The first boundary shows both CTCF and Pol II enrichment, while the second one show Cp190 and Pol II enrichment. Interestingly, BEAF-32 is not enriched although it has been proposed as one of the most enriched architectural protein at TAD borders in S2 cells (<https://www.nature.com/articles/s41467-017-02526-9>). CTCF has previously been proposed not to be enriched at TAD borders in *Drosophila* embryo or embryo derived cells (<https://www.nature.com/articles/s41467-017-02525-w>; <https://www.nature.com/articles/s41467-017-02526-9>;

<https://www.sciencedirect.com/science/article/pii/S0092867417303434>), but has been found to be enriched in a BG3 cells (derived from the larval CNS)

(<https://genome.cshlp.org/content/29/4/613.short>). Most importantly the authors show that removal of this boundary leads to reorganisation of the domains indicating that CTCF would have a functional role in 3D chromatin organisation in *Drosophila*. Finally, deleting the second boundary between the two domains that span over Notch leads to loss of a mega domain and down-regulation of approximately 2/3 of the genes in that mega domain. This is important because it provides evidence that the 3D chromatin organisation has a role in gene regulation. The results in this manuscript contrast other recent studies (e.g., <https://www.nature.com/articles/s41588-019-0462-3>), suggesting that proper dissection of some TAD borders can indeed shed light on the functional role of 3D chromatin organisation and that the domain boundaries are essential for correct gene regulation.

The paper is well written and presents very important results. There are some points that authors would need to address before I could recommend this paper for publication:

1. The authors claim the two-domain organisation of the Notch locus is present in all cells. We checked this in BG3 cells and Kc167 cells (data from <https://genome.cshlp.org/content/29/4/613.short>) and found that this is not true for all cells. While in BG3 cells, this seems to be the case, Kc167 cells have a different organisation (see attached figure).

2. Lines 269-272. I only observed that for 183 mutant and not for the other mutants. The figure needs to be better labeled and quantifications of the changes compared to WT need to be added on the plot so we can understand what the authors are referring to.

3. Figure 5b is also not clearly labeled and needs the quantifications added. Where is the viewpoint? Is the mutant data presented in the second row? Etc.
4. For the differentially expressed genes (figure 5C), I would like to see the logFC and p-value volcano plot of all genes in the locus, just to give a better representation of the argument the authors try to make.
5. Line 457-459. The authors talk about this being conserved in human cells and cite a manuscript in preparation. Since we cannot see that data, I would suggest they either remove the sentence or provide a preprint version of the manuscript so we can verify whether the statement is correct or not.
6. Lines 585-586. In the methods section, the authors mention they computed the correlation between the biological replicates, but they do not provide any parameters they used to compute that. In particular, it is essential to at least know the bin size they used.
7. Figure 5. We had a look at whether the Notch enhancer mega-domain is present in other cells and observed that this is not the case (see attached Figure). Can the authors comment why this locus would show different organisation in different cells.

Reviewer #3 (Remarks to the Author):

The manuscript of Arzate-Mejia and colleagues explores the role of genome features in topological domains and the impact of these on transcription. They focus on the Notch genomic region as a test case to map topologically associating domains (TADs). Using *Drosophila* S2R+ cell culture systems, they remove regional boundaries and assess the effects on TAD structure and transcription. They take advantage of previously published data during embryogenesis to examine the timing of Notch TAD formation relative to zygotic transcription and argue that RNA Pol II binding, but not transcription is important for TAD boundary establishment. In addition the binding sites of known architectural proteins are shown to be required at the Notch locus for normal chromosome interactions read out as "domain structure".

While the authors provide a nice test case for how boundary elements and RNA Pol II binding regulate TADs and the impact this has on transcription, I feel that the paper should go further in understanding at least some additional aspects: to what extent does RNA Pol II binding correlate throughout the genome with TAD boundaries? What is the nature of the "putative exonic promoter" of Notch? Also, while likely beyond the scope of the manuscript, understanding how these mutations impact *in vivo* biology will be also important in the long term.

1. To make the conclusion about RNA Pol II binding more generalizable, can the authors examine this relationship more globally?
2. In particular, the RNA Pol II binding in exon 6 of Notch has not been previously noted, seems very unusual, and should be better investigated. To my knowledge this has not been described *in vivo* and raises the question as to whether S2R+ cells have an unusual regulation of the Notch

gene. Is it truly a promoter? What RNA species would this make? Are there RNAs produced in both directions or only towards the 3' of Notch? Would it be predicted to be translated? If so, what portion of Notch does this correspond to? Is there a more global relationship with TAD boundaries and intragenic Pol II binding sites/ or alternative promoters? A better investigation here is merited and then the authors can either call it a promoter or not. As it is in the text, they oscillate between "putative promoter" to "intragenic promoter".

3. The transposonable elements in this region are ignored in the manuscript. I feel it is important to include these on the genome maps, annotated from their DNaseq data. Those of S2R+ cells may differ widely from the reference genome. Given the recent suggestion in mammalian cells that retrotransposons may be important boundary elements, these sequences should be positioned on the coordinates. Where are TEs positioned in reference to the numerous promoter, enhancer and AP binding regions?

4. There seem to be differences in Notch TADs and Pol II binding in Kc cells from Ramirez F, Nat Comm. Can the authors comment on these differences?

5. On p7 they state that the establishment of boundaries at Notch strongly correlates with the progressive gain of chromatin accessibility and the binding of Zelda and RNA-Pol II
I do not see the evidence for Pol II at the "putative internal promoter" in these data. Why is this?

6. It is unclear in the methods if the CRISPR cell lines are from single cell clones or not. Are they working with mixed populations of cells or not? Additionally, I believe that S2R+ cells are male, but tetraploid. Therefore there are 2 X chromosomes- are they both mutated in the same way?

7. Many of the Figures need to be enlarged and with better resolution: Sup 1c and sup 2b for example. Also a lot of important data are in Sup Fig 5A, which is difficult to see since the resolution is bad. It would help to separately letter each subfigure so it can be referenced separately in the text.

8. It might be helpful to plot some of the data as log fold changes (Figure 1B for example), which would make the significant changes in CTCF of K27me3 more visible.

- **Answers for Reviewer 1**

1. Figure 1b. Is it known whether the “main” and exonic promoters of *Notch* are transcribed in the cells used for these analyses? It is interesting that Domain 1 lacks H3K27ac, Pol II, and H3K4me3, while Domain 2 contains all these marks. It would be interesting to show the distribution of H3K27me3, even if it is from a related cell line. The presence of H3K7me3 in domain 1 but not domain 2 would suggest that the two domains are formed by their different transcriptional states.

We appreciate the reviewer’s observations and provide additional data to try to address her/his questions.

In Figure 1a of our manuscript we show ChIP-seq tracks for APs, histone post-translational modifications and RNA Pol II at the *Notch* locus. We observe two regions with marked enrichment for RNA Pol II binding, located at the 5’ end of *Notch* and at exon 6 (Figure 1a) and validated such binding in S2R+ by qChIP (Figure 1b). To investigate if both RNA Pol II bound-regions are transcribed, we analysed publicly available GRO-seq data and used processed START-seq data all from S2 cells from which S2R+ cells were derived and complemented the analysis with our RNA-seq dataset for S2R+ cells. We observe that all exons at *Notch* are transcribed, although at different levels, with exons located in domain 2 being transcribed at higher levels (see Figure R1-1 bottom panel – total RNA-seq). Also, and as expected, we observe that the promoter of *Notch* is transcribed, based both on GRO-seq and START-seq data (Figure R1-1). Interestingly, we also observe nascent transcription at the site of RNA Pol II enrichment on exon 6 (Figure R1-1). These data suggest that at the *Notch* locus both genomic regions occupied by RNA Pol II are transcribed based on nascent RNA datasets (Figure R1-1).

Regarding the presence of H3K27me3 histone modification at the *Notch* locus, we observe that this mark is generally not enriched (Figure R1-1). To further validate this observation we obtained a processed signal file from a ChIP-chip experiment against H3K27me3 in S2 cells from the modENCODE project. As shown in Figure R1-2, the histone modification H3K27me3 is mostly depleted from the locus in S2 cells, consistent with the ChIP-seq data (Figure R1-1) and with our qPCR results for the 5’ end of *Notch*, the B2 boundary and exon 6 (Figure 1b).

Therefore, in S2/S2R+ cells both *Notch* contact domains contain “active” histone post-translational modifications and the exons within each domain are transcribed. However, we observe differences in both histone mark enrichment and transcription levels between domains, with domain 2 presenting higher levels of both, which could influence *Notch* 3D organization. Importantly and as mentioned, the “repressive” histone post-translational modification H3K27me3 is not enriched at the *Notch* locus. Then, as stated in the discussion of our manuscript and based on the data derived from CRISPR mutants we propose that at least two mechanisms are at play organizing the *Notch* locus. One dependent on genetic sequences that recruit APs and RNA Pol II and one that is driven by the chromatin status of the sequences.

Figure R1-1

Figure R1-1. The chromatin landscape and 3D organization of the *Notch* locus in S2R+ cells.

Figure R1-1. Chromatin landscape and 3D organization of the *Notch* locus in S2R+ cells. This figure is related to Reviewer 1 question 1. Shown is a heatmap of Hi-C data a 1k resolution from S2R+ cells at the *Notch* locus along ChIP-seq tracks for ATAC-seq, histone post-translational modifications, STARR-seq, START-seq, GRO-seq and RNA-seq. Highlighted

are regions at which ATAC-seq signal is enriched for “active” histone modifications along the *Notch* locus. Two of those regions show enhancer activity in S2 cells according to STARR-seq data (purple tracks below STARR-seq data track).

Figure R1-2

Figure R1-2. The histone post-translational modification H3K27me3 is depleted at *Notch* in S2 cells.

Figure R1-2. The histone post-translational modification H3K27me3 is depleted at *Notch* in S2 cells. This figure is related to Reviewer 1 question 1. Shown is a ChIP-chip signal track for H3K27me3 in S2 cells at the *Notch* locus.

2. “data suggest that the Promoter Proximal Region of *Notch* is necessary to promote intra-TAD interactions at D1 domain and to restrict communication between the 5’UTR with the neighboring TAD”. Authors should be careful with this type of interpretation. If I cut my hand off, for example, and I start bleeding profusely from the residual arm, this result does not mean that the normal role of the hand is to restrict bleeding from the arm. The role of the promoter-proximal region of *Notch* is to recruit proteins necessary for transcription.

We agree with the reviewer. We have removed this sentence from the manuscript.

From the manuscript:

“The interaction profile of the 5’UTR of *Notch*, as evaluated by Virtual 4-C, shows a decrease in interactions with the D1 domain concomitant with a gain of ectopic interactions with the upstream *kirre*-domain 2 (Supplementary Fig. 5b). However, this trend is not observed when additional viewpoints within the D1 domain are analyzed (Fig. 2c). This data suggest that the Promoter Proximal Region of *Notch* is necessary to promote intra-TAD interactions at D1 domain and to restrict communication between the 5’UTR with the neighboring TAD. Despite observed topological effects, the removal of the PPR is not sufficient to induce the fusion of the D1 domain and the upstream *kirre*-domain-2.”

3. “The increase in inter-domain interactions is accompanied by a decrease in the number of interactions within each domain, likely reflecting the engagement of their sequences in ectopic contacts (Fig. 2d; Supplementary Fig. 5a)”. This is certainly possible but, more likely, since the comparison between the two samples requires normalizing for the number of reads, if something goes up something else has to come down, no matter what.

Although we agree with the reviewer on the effect of normalization by the number of reads, we observe reciprocal changes in interactions (gain of inter-domain interactions accompanied by loss of intra-domain interactions) even in the absence of normalization by the number of reads. In Figure R1-3a we show heatmaps at 5kb resolution for a 750 kb region centred in *Notch* for the wild-type and the *5pN-Δ183* mutant. For the mutant *5pN-Δ183* we observe the reported increase in inter-domain interactions between the *kirre* and *dnc* loci and a decrease in the frequency of interactions within domains as compared to wild-type in both normalized and non-normalized data sets. Also, we show Virtual-4C profiles for only non-normalized data using a region inside the *kirre* and the *dnc* domains as viewpoints. In both, an increase in interactions between domains is evident (Figure R1-3b).

Figure R1-3

Figure R1-3. Changes in inter-domain and intra-domain interactions upon B1 boundary disruption are observed in data non-normalized by read number.

Figure R1-3. Changes in inter-domain and intra-domain interactions upon B1 boundary disruption are observed in data non-normalized by read number. This figure is related to Reviewer 1 question 3. a, Heatmaps of Hi-C data at 5 kb resolution for wild-type and the 5pN-183 mutant no normalized by read number (left) or normalized by read number (right). b, Virtual 4C for wild-type and the 5pN-183 mutant using anchors inside *kirre* and *dnc* using data non-normalized by read-number.

4 “Moreover, visual inspection of Hi-C heatmaps at 5 kb resolution suggest that inter-domain interactions are constrained by the 5’ and 3’ boundaries of the *kirre* domain-1 and the *dnc* domain, respectively (Fig. 2d)”. Here and throughout the manuscript, authors assume that domains are formed by a boundary that does not allow interactions to occur between the two

adjacent domains. This is what happens between two adjacent CTCF loops in mammals, where continuous extrusion by cohesin may interfere with interactions between the two loops. However, it is unclear whether this is also the case in *Drosophila*. More likely, the boundary is formed as a consequence of sequences on either side of the boundary binding to proteins that, because of their biochemistry, prefer to interact with other proteins present on sequences located either to the left or the right of the boundary. Therefore, it is not that there is a barrier to interactions. Rather, proteins actively determine the direction of interactions based on the location of their chemically preferred partners. If, as the authors suggest, architectural proteins form a barrier to interactions, what would be the mechanism by which this happens?

We appreciate the reviewer's comments and also agree that in *Drosophila* boundaries are likely formed due to preferential interactions between proteins at the domain boundaries. For example, CP190, a protein present at most of the domain boundaries, can mediate long-range interactions between different genomic sequences due to direct dimerization of BTB domains present in the protein (Liang et al., 2014; Vogelmann et al., 2014). A similar function has also been proposed for other proteins present at domain boundaries like Chromator (Vogelmann et al., 2014) and more recently to Peapkskeak (Gutierrez-Perez et al., 2019). Also, there are evidences indicating that CTCF can form dimers in an RNA dependent manner, although these evidences come from studies in mammalian systems.

Then, at *Notch*, domain boundaries interact likely due to the presence of proteins like CP190 and CTCF among others. Removal of the B1 boundary possibly results in a reconfiguration of preferred interactions between sequences occupied by APs like CP190 and the resulting set of long-range contacts, is what we observe as new interactions taking place between the 5' boundary of *kirre-1* and the 3' boundary of *dnc*.

To avoid confusion we replaced the sentence:

"Moreover, visual inspection of Hi-C heatmaps at 5 kb resolution suggest that inter-domain interactions are constrained by the 5' and 3' boundaries of the *kirre* domain-1 and the *dnc* domain, respectively (Fig. 2d)" for:

"Moreover, visual inspection of Hi-C heatmaps at 5 kb resolution suggest that the 5' and 3' boundaries of the *kirre* domain-1 and the *dnc* domain, engage in new long-range interactions likely reflecting a preference of the protein complexes at each boundary to physically interact (Fig. 2d)"

5. Although the sites for architectural proteins appear to be together based on Figure 1A, there are clear differences between the locations of the peak summits. For example, GAF does not seem to overlap with SuHw or CTCF. It may help in the interpretation of the phenotypes if the authors show a figure panel, similar to Figure 1D, showing the deleted sequences and the location of the binding motifs of the different architectural proteins, which should be located at the summits of the peaks. This will complement results in Figure 2F and also allow the authors to interpret the deletions in the context of the different architectural proteins affected. This information may be important in the interpretation of the results. For example, D102 appears to delete the site for M1BP but not CTCF, D183 deletes the site for CTCF but not M1BP, and D343 appears to delete both.

We appreciate the reviewer suggestions and agree that there are differences in the peak summits for Architectural Proteins, in particular for the GAGA Associated Factor (GAF). In Figure 1d we included just the location of DNA binding motifs detected by JASPAR (p -value < 0.0001). In order to provide

more information about binding sites for APs at the 5' region of *Notch* we have included an ATAC-seq track for S2 at Figure 1d and also a track with the peak summits identified by MACS2 as shown (Figure R1-4).

Figure R1-4

Figure R1-4. CRISPR mutants for the B1 boundary of *Notch* remove binding sites for APs.

Figure R1-4. CRISPR mutants for the B1 boundary of *Notch* remove binding sites for APs. This figure is related to Reviewer 1 question 5. Shown is a 1.5 kb region between *kirre* and *Notch*. Motif binding sites identified by FIMO (p-value < 0.0001) and peak summits identified from ChIP-seq data by MACS2 are also shown as well as the regions deleted by CRISPR in each mutant clone. This figure has been added to Figure 1 as Figure 1d.

Furthermore we have added three lines in the main manuscript highlighting the motifs and peak summits removed by all our deletions as follows:

“Deletion of the Promoter Proximal Region of *Notch* (PPR; *5pN-Δ102*) removes M1BP DNA binding site but not CTCF DNA binding site as well as the ChIP-seq peak summits of CTCF, Su(Hw), Pita and BEAF-32 (Figure 1d)...”

“Deletion of the Intergenic Region (IR; *5pN-Δ183*) upstream of the PPR of *Notch* removes the CTCF DNA binding site but not M1BP as well as the ChIP-seq peak summits of CTCF, Ibf1 and Ibf2 (Figure 1d)...”

“We hypothesized that deletion of both sequences, thus removing the DNA binding motifs of M1BP and CTCF as well as all ChIP-seq peak summits for APs at the region...”

We believe that the inclusion of this information will aid in the interpretation of the results.

6. “A similar topological effect is observed with the 5pN-Δ755 mutant, although less severe (Fig. 2a; Supplementary Fig. 5a).” This deletion affects a larger region of the genome than D343, but the effect is less severe. One possibility would be that sequences located between the right breakpoints of D755 and D343 i.e. sequences containing the binding site for M1BP and potentially other proteins are responsible for the increased effect of D343. As suggested in comment #7, paying attention to the specific binding sites deleted by the various deletions may be important in the interpretation.

We agree with the reviewer that the presence of M1BP and likely other motif sequences could be responsible for the mild effect observed for the mutant *5pN-Δ755*. We have include a line in the main manuscript:

“A similar topological effect is observed with the *5pN-Δ755* mutant, although less severe, likely reflecting the presence of the M1BP motif and of other APs (Fig. 2a; Supplementary Fig. 5a)”

7. “The marked decrease in RNA Pol II binding at the B1 boundary in the 5pN-Δ183 mutant suggest that the 5’ intergenic region of *Notch* is essential for RNA Pol II recruitment”. The intergenic region is larger than the region covered by D183. The results only suggest that sequences contained within the D183 deletion are important for Pol II recruitment.

We completely agree with the reviewer. We have changed the sentence in the manuscript for:

“The marked decrease in RNA Pol II binding at the B1 boundary in the *5pN-Δ183* mutant suggest that sequences within the 183bp region are important for RNA Pol II recruitment”

8. Figure S5D. It is unclear how the authors can conclude that the data in this figure indicate that CTCF is specifically required to recruit Pol II.

We appreciate the reviewer observation. It was not our intention to transmit the idea that CTCF recruits directly RNA Pol II based on the data derived from the EMSA experiment shown in Figure S5D. Instead, our results suggest that mutation of the CTCF binding motif affects (either directly or indirectly)

RNA Pol II recruitment at this region. In order to clarify we have changed the main manuscript as follows:

Then, in this case, the CTCF DNA binding motif seems important to either directly or indirectly recruit RNA Pol II. Moreover, mutation of the CTCF motif results in loss of nuclear proteins binding (Supplementary Fig. 5d)

9. Figure 5A. “Visual inspection of wild-type Hi-C data binned at 20 kb resolution reveals the presence of 1Mb sized Topologically Associated Domain with boundaries overlapping the Notch intronic enhancer and a ~100 kb gene desert downstream of Notch (Fig. 5a,b)”. I disagree with this interpretation. What the authors see is a compartmental interaction between sequences in Notch and a small domain located on top of the “0” of the “20 kb res” legend of the heatmap. A distance-normalized heatmap or the Pearson correlation will show that the signal corresponds to a compartmental interaction.

We appreciate the reviewer’s observations and provide additional analysis to address his/her concerns. In agreement with the reviewer, a distance normalized Pearson correlation heatmap at 10 kb resolution shows that *Notch* and the downstream ~100 kb gene desert are located within a larger compartmental interaction between two B compartments (chrX:2,760,000-3,340,000 and chrX:3,980,000-4,080,000) (Figure R1-5a,b and Figure 5a,b). However, we reasoned that as the upstream region of *Notch* is also in the B compartment, we would expect a similar interaction frequency between this region and the gene desert if this contact was purely the result of an interaction between compartments, but this does not seem to be the case (see Figure R1-5a). Indeed, at 10 kb resolution we observe two bins from *Notch* Domain 1 that show high-frequency of interactions with the gene desert. In contrast the frequency of interactions of bins located upstream of *Notch*, which also belong to the same B compartment, are lower (see Figure R1-5a). Thus, the frequency of the detected contacts between the *Notch* locus and the gene desert are higher than the compartment interactions spanning the region. Therefore, even though the interacting sequences forming the Megadomain belong to the same compartment, the interactions between them are stronger than the overall compartment contacts in the region. Although we do not discard a contribution of compartmental interactions to the formation of the Megadomain our data suggest the sequences at the *Notch* B2 boundary are crucial for establishing these high-affinity long-range interactions between the *Notch* D1 domain and the gene desert, both at B compartments, as upon its genetic removal, the contacts are lost while the compartments at the anchors remain largely unaffected (See Figure 5a, and new virtual 4C analysis in Figure R1-5c).

Then, for clarity and attending the reviewer’s observations we have modified the sentence:

“Visual inspection of wild-type Hi-C data binned at 20 kb resolution reveals the presence of 1Mb sized Topologically Associated Domain with boundaries overlapping Notch intronic enhancer and a ~100 kb gene desert downstream of Notch (Fig. 5a,b)”

As follows:

Visual inspection of wild-type Hi-C data binned at 20 kb resolution reveals the presence of a 1Mb sized contact domain with anchors overlapping the *Notch* B2 boundary and a ~100 kb gene desert downstream of *Notch* both located within chromatin compartment B (Fig. 5a,b)”

Figure R1-5. The *Notch* locus interacts with a gene desert ~1Mb downstream. This figure is related to Reviewer's 1 question 9. a, A heatmap of 10kb resolution from wild-type Hi-C data show a long-range interaction between *Notch* and a gene desert. Shown is the PC1 at 10 kb resolution. A rectangle highlights a compartmental interaction between B compartments. b, Distance normalized Pearson correlation heatmap at 10 kb resolution from wild-type Hi-C data. Red and blue indicate positive and negative correlation, respectively. Shown are ChIP-seq tracks for histone modifications and a PC1 track. A zoom-in is shown at the right. A rectangle highlights the region in the heatmap corresponding to the gene desert. c, Virtual 4C using wild-type and mut-enh Hi-C data using different viewpoints along the *Notch* locus. See that the enhancer at the B2 boundary mediates the long-range interaction between *Notch* and the gene desert.

10. Figure 5A. “domain boundaries are engaged in a long-range interaction similar to what has been observed for anchors of chromatin loops mediated by Pc and by CTCF”. Interactions mediated by CTCF in mammals or Pc in Drosophila appear as circular dots of signal with an intense central pixel surrounded by pixels of decreasing intensity. The interaction marked by an arrow in Figure 5a does not look like this. It has the shape of a line, indicating interactions between a short sequence in *Notch* and a large domain to the right of the map. A rough calculation suggests this domain is 118 kb.

We agree with the reviewer and we have provided a detailed answer in 9. For clarity we have removed the sentence:

“First domain boundaries are engaged in a long-range interaction similar to what has been observed for anchors of chromatin loops mediated by Pc and by CTCF”

11. “although we observe multiple sub-TADs at Nenh344 -mega-domain”. At 1 kb resolution these domains were referred to as TADs. Now, at 20 kb resolution, they are called sub TADs. What determines if a domain is a TAD or a subTAD?

To avoid confusion we will refer to the triangles in the diagonal as TADs irrespective of resolution. We have changed the sentence:

“although we observe multiple sub-TADs at Nenh-mega-domain”

for:

“although we observe multiple TADs at Nenh-mega-domain”

12. “The overall organization of the Nenh346 -mega-domain is similar to loop domains described in mammals, where regions at the loop anchor display a high frequency of interaction while at the same time sequences within the domain interact”. As described in comment #10, I disagree with this interpretation. Loop domains in vertebrates are formed by cohesin extrusion and their anchors contain CTCF, which binds to a short sequence of a few bp, rather than a 118 kb domain. This large domain would normally be classified as a compartment.

We agree with the reviewer and we have provided a detailed answer in 9. For clarity we have removed the sentence:

“The overall organization of the Nenh346 -mega-domain is similar to loop domains described in mammals, where regions at the loop anchor display a high frequency of interaction while at the same time sequences within the domain interact” from the main manuscript.

13. **“Together, our data show that the enhancer at the intron five of Notch is a topological insulator that restricts communication between the two topological domains of Notch. Also, it is essential to recruit RNA Pol II at Notch promoters and therefore for Notch transcription”. The enhancer of Notch is in an intron and appears to contact both the upstream and downstream promoters. These directional contacts create two small domains on both sides of the enhancer. Interpreting these results to say that the enhancer is “a topological insulator that restricts communication between the two topological domains” is inappropriate. If enhancers are insulators, then we don’t need two different words to name these sequences.**

We agree with the reviewer’s comments and have replaced the phrase:

“Together, our data show that the enhancer at the intron five of Notch is a topological insulator that restricts communication between the two topological domains of Notch”

for:

“Together our data show that upon enhancer loss contacts are gained between the two Notch domains. Also the enhancer **sequence seems essential to recruit...”**

14. **Figure 5B shows a virtual 4C analysis of interactions between the Notch enhancer and the downstream domain shaded in purple. Are all the genes whose expression is affected by the deletion of the enhancer within this purple region? In Figure 5A it appears that the enhancer contacts many sequences in what the authors call mega domain, not just the ones shaded in purple in Figure 5B. Are these other interactions not statistically significant? Authors should comment on how the enhancer affects the transcription of genes located between the enhancer and the purple shaded domain in Figure 5B i.e. those that appear not be contacted based on the virtual 4C.**

We appreciate the reviewer’s comments. The genes located in the purple shaded region in Figure 5B are the ones located at the distal end of the Mega domain in the gene *desert*. The deletion of the Notch enhancer results in changes in the expression of genes located along the megadomain, both inside and at the anchors (Figure 5c and Figure R1 6-b). The Notch enhancer establishes significant contacts with sequences at the gene *desert* as analysed at the restriction fragment level by a virtual 4C (see Methods). At this resolution, the contacts between the enhancer and regions within the megadomain do not pass the statistical threshold. However, as observed in Figure 5a, at 20kb resolution there are significant interactions between the Notch locus and the regions within the megadomain that are lost upon enhancer deletion. Interestingly, we observe that while downregulated genes located at the anchors interact with the Notch locus (Supplemental Figure 6), those located inside the megadomain that also become downregulated (Figure 5d and Figure R1 6-b), also lose interactions with neighbouring sequences within the domain, that are enriched for histone marks associated with active enhancers (Figure R1-6a,b).

Figure R1-6

Figure R1-6. Deletion of the *Notch* enhancer disrupts the Megadomain and affects gene transcription.

Figure R1-6. Deletion of the *Notch* enhancer disrupts the Megadomain and affects gene transcription. This figure is related to Reviewer 1 question 14. a, Virtual 4C using wild-type and mutant-enh Hi-C datasets and as viewpoints loci that became downregulated upon enhancer deletion. Highlighted are regions with changes in interaction in the enhancer-mutant. b, RNA-seq tracks for wild-type and enhancer mutant for the genes used as viewpoints for the Virtual 4C analysis in a.

Minor comments

1. I'm not familiar with the term "in nucleus Hi-C". Authors should explain how this Hi-C method differs from the commonly used in situ Hi-C.

The term "in nucleus HiC" comes from the original paper by Nagano et al., in which the protocol was described for the first time in the context of the single cell HiC experiment (Nagano et al., 2013). Afterwards, Rao et al published an equivalent protocol and called it "in situ HiC" (Rao et al., 2014).

In the first paragraph of the results section from Rao et al. it reads: "Our in situ Hi-C protocol involves crosslinking cells with formaldehyde, permeabilizing them with nuclei intact, digesting DNA with a suitable 4-cutter restriction enzyme (such as MboI), filling the 5-overhangs while incorporating a biotinylated nucleotide, ligating the resulting blunt-end fragments, shearing the DNA, capturing the biotinylated ligation

junctions with streptavidin beads, and analyzing the resulting fragments with paired-end sequencing (Figure 1A). **This protocol resembles a recently published single-cell Hi-C protocol (Nagano et al., 2013), which also performed DNA-DNA proximity ligation inside nuclei to study nuclear architecture in individual cells.** The Rao et al. authors however, do not explain what is new and different in their in situ HiC vs the in nucleus HiC protocols to name it differently, and upon comparison, we found there are no significant differences. Thus, we go for the name of the original protocol, in nucleus HiC (Nagano et al., 2013; Nagano et al., 2015).

2. Authors use the term “topological insulator”, which I have not seen used in this context before. Is this a new type of insulator, different from those described before? Wikipedia says, “A topological insulator is a material with non-trivial symmetry-protected topological order that behaves as an insulator in its interior but whose surface contains conducting states, meaning that electrons can only move along the surface of the material”. Authors should make sure this is what they mean by topological insulator. In general, the authors over-use the term “topological” throughout the manuscript. For example, in these two consecutive sentences “As shown, loss of the B1 boundary does not affect the topological organization of the D2 domain of Notch, which suggests that it is an independent topological unit. The topological organization of the D2 domain could result from the activity of a topological insulator” the authors use “topological” 4 times. I would just use delete every instance of this word in the manuscript.

We appreciate the reviewers comment and correction. We have replaced “topological insulator” for the more standard “chromatin insulator” and reduce the use of topological across the manuscript.

3. Figure S5C. It is not clear what the title “CTCF en AbdB” in the figure panel means.

We have modified it for “CTCF at AbdB locus”.

- **Answers for Reviewer 2**

1. The authors claim the two-domain organization of the *Notch* locus is present in all cells. We checked this in BG3 cells and Kc167 cells (data from <https://genome.cshlp.org/content/29/4/613.short>) and found that this is not true for all cells. While in BG3 cells, this seems to be the case, Kc167 cells have a different organization (see attached figure).

We appreciate the reviewer's observation and agree that *Notch* shows a different three-dimensional organization between the Kc167 cell line and the S2R+ and BG3 cell lines. In that regard, it has come to our attention that Kc167 cells are of female origin while the S2R+ cells and BG3 cells are of male origin (Lee et al., 2014). Since *Notch* is on the X chromosome and it is subject to dosage compensation in male cells, it is possible that topological differences between female and male cell lines at *Notch* could reflect sex specific regulation. Also, since the three cell lines correspond to different stages of development we cannot discard specific conformational changes due to cell of origin as has been recently reported between the Kc167 and BG3 cell lines (Chathoth and Zabet, 2019). Then, to provide a comprehensive picture of the 3D organization of *Notch* in *Drosophila* we have re-analyzed Hi-C datasets derived from the female embryonic cell line Kc167 (Cubañas-Potts et al., 2017; Li et al., 2015) and the male L3 cell line BG3 (Chathoth and Zabet, 2019) (Figure R2-1) and compared *Notch* 3D organization with the one observed in S2R+ cells (this work) and in early embryos. Also, and when available, we analyzed the chromatin structure and the binding of APs at the *Notch* locus in the given cell lines.

Analysis of Hi-C data during early embryogenesis, shows that *Notch* becomes progressively organized in the three dimensional nuclear space (Figure R2-1a). At nuclear cycle 13 (nc13) *Notch* appears to be organized into a single domain of interaction demarcated at the 5' end by the B1 boundary (5' end of *Notch*) while at nuclear cycle 14 (nc14), and coincident with the zygotic genome activation, a two domain organization is readily evident which concurs with the detection of the B2 boundary at the intron 5 of *Notch* (Supplementary figure 2 and Figure R2-1). Importantly, the appearance of domain boundaries at the *Notch* locus correlate with the acquisition of chromatin accessibility and binding of RNA Pol II, TBP and Zelda at both sites (although with marked differences in enrichment at each boundary) (Supplementary figure 2b,c). By stage 3-4hrs the two-domain organization of *Notch* is fully evident and this correlates with high transcriptional levels as reported by modENCODE for that stage of development (<http://flybase.org/reports/FBgn0004647#expression> and Fig R3-2h), suggesting that the two domain organization of *Notch* is partly linked to transcriptional activation. Of note, the above observations result from analysis of Hi-C data from mixed male and female embryos, thus a sex derived structural difference cannot be distinguished.

In agreement with the reviewer, we observe a single domain spanning the *Notch* locus in the female derived Kc167 cell line in contrast with the two domain organization evident in the male derived cell lines S2R+ and BG3 which suggest that the different 3D organization at the *Notch* locus could be related to dosage compensation between female and male cells. In *Drosophila*, dosage compensation at active loci in male cells occurs due to the recruitment of the MSL complex which results in increased transcription and the deposition of the histone post-translational modification H4K16ac into the gene bodies of dosage compensated loci (Lucchesi and Kuroda, 2015). To further relate the presence of this histone mark with differences in genome organization at the *Notch* locus we obtained processed signal files from modENCODE for H4K16ac available for the three cell lines. We observe a clear difference in

the enrichment of H4K16ac between the female cell line (Kc167) and the male cell lines (S2R+ and BG3), with H4K16ac being highly enriched at the *Notch* region encompassing the Domain 2 detected in male cells (exon 6 – exon 9) while the histone post-translational modification H3K27me3 is enriched at the same region in female cells and overall depleted in male cells (Figure R2-1b and Figure R1-1). Furthermore, the differential enrichment of H4K16ac and the two domain organization of *Notch* correlates with higher expression levels in male cells (see modENCODE expression RPKM per cell line Figure R2-1b) overall suggesting that higher transcription and the enrichment of histone marks like H4K16ac correlate with the presence of the two domain organization of *Notch* in male cell lines.

In addition to histone post-translational modifications, the activity of domain boundaries at *Notch* locus could also result in differences in 3D conformation. As supported by our results, the B2 boundary is important for *Notch* 3D organization into two domains, as its removal results in ectopic interactions between the two domains of *Notch* resulting in a single interaction domain (Figure 4b), a 3D conformation similar to the one observed in Kc167 cells. To relate the presence of APs at the B2 boundary with the two domain organization of *Notch* we re-analyzed publicly available ChIP-seq datasets for Kc167 cells and compared their binding profile at *Notch* with the one observed in S2/S2R+ cells, as both cell lines are of embryonic origin, which allow us to make a more direct comparison. We observe differences in APs occupancy at the B2 boundary between both cell types in contrast with the APs occupancy at the B1 boundary (Figure R2-1c). While the Boundary 1 is occupied by 8 DNA binding APs, CP190 and RNA Pol II in both cell lines, the B2 boundary shows an overall reduction in the binding of APs like RNA Pol II, CP190, ZIPIC, Su(Hw) and CTCF in the Kc167 cell line. Importantly, the observed difference in APs binding at the B2 boundary genomic region in Kc167 cells is not due to differential chromatin accessibility since Kc167 and S2 cells show a remarkable similar ATAC-seq profile at the *Notch* locus. For example, CP190 is a protein with important roles in insulator activity and reduction of CP190 can result in disruption of long-range interactions and changes in gene expression (Liang et al., 2014; Ong et al., 2013; Wood et al., 2011). Since CP190 cannot bind directly to DNA, the recruitment of CP190 could be modulated by differential binding of APs with DNA binding capacity that directly interact with CP190, like CTCF or Su(Hw) (Figure R2-1c, right). It is important to point out that direct comparison between different sets of ChIP-seq experiments performed by different laboratories could lead to differences due to technical aspects and therefore conclusions about differential binding should be taken cautiously.

Therefore, the differential 3D organization detected at the *Notch* locus, which undergoes dosage compensation in males, correlates with differences in transcription, the enrichment of histone modifications like H4K16ac and the binding of APs. The dynamic organization of this locus contrast with early observations suggesting a mostly invariant organization of the genome in *Drosophila* and supports recent observations that boundaries are dynamic between cell types and that these variability correlates with the binding of APs like CTCF (Chathoth and Zabet, 2019).

Based on the presented data we have modified the phrase in the manuscript:

“A heatmap of Hi-C data binned at 1 kb resolution shows that *Notch* is organized into two topological domains (Fig. 1a)” for:

“A heatmap of Hi-C data binned at 1 kb resolution shows that *Notch* is organized into two topological domains in S2R+ cells (Fig. 1a)”

We have also modified the section’s title and replaced it for:

“*Notch* is organized into two topological domains in S2R+ cells”

Figure R2-1

Figure R2-1. The 3D organization of the *Notch* locus in *D. melanogaster*

Figure R2-1. The 3D organization of the *Notch* locus in *D. melanogaster*. This figure is related to Reviewer 2 question 1. a, Heatmaps at 1 kb resolution centered in Notch of Hi-C data derived from different stages of early embryonic development of *D. melanogaster*. b, Heatmaps at 1 kb resolution centered in Notch of Hi-C data derived from different *D. melanogaster* cell lines. Next to each heatmap is shown the expression level of Notch at the indicated cell line as obtained from modENCODE. Below each heatmap are shown ChIP-chip tracks for H4K16ac and H3K27me3 as obtained from modENCODE. c, Chromatin accessibility and binding of different APs and RNA Pol II at *Notch* domain boundaries in Kc and S2/S2R+ cells.

2. Lines 269-272. I only observed that for 183 mutant and not for the other mutants. The figure needs to be better labeled and quantifications of the changes compared to WT need to be added on the plot so we can understand what the authors are referring to.

We agree with the reviewer and we have modified the manuscript as follows:

Original lines 269-272:

“(Fig. 3d). In contrast, the transcript levels at the D2 domain remain either unaffected (mutant 5pN- Δ 343) or show a tendency towards an increase (mutant 5pN- Δ 102, Δ 183, and Δ 755) (Fig. 3b) which correlates with a gain in contacts between the exonic promoter of Notch with the D2 domain (Fig. 3e).”

for:

“(Fig. 3d). In contrast, the transcript levels at the D2 domain remain either unaffected (mutant 5pN- Δ 343) or show an increase (mutant, Δ 183) (Fig. 3b). In particular the increased transcription at the 5pN- Δ 183 correlates with a gain of interactions between sequences at the D2 domain (Fig.3e)”

Also we have added more information on the Figure legend for Figure 3d as follows:

Original line 952:

“d Virtual-4C for wild-type and mutant clones using the 5' UTR (top) and the exonic promoter of Notch (bottom) as viewpoints. Shown are the percent of interactions between the viewpoints and the kirre domain-2, the Notch D1 domain, the Notch D2 domain and the dnc domain. ChIP-seq tracks for CP190 and RNA Pol II are also shown”

for:

“d Virtual-4C for wild-type and mutant clones using the 5' UTR (top) and the exon 6 (bottom) as viewpoints. Shown are the percentage of interactions between the viewpoints and regions within the kirre domain-2, the Notch D1 domain, the Notch D2 domain and the dnc domain for both wt and all mutant samples. ChIP-seq tracks for CP190 and RNA Pol II are also shown”

3. Figure 5b is also not clearly labelled and needs the quantifications added. Where is the viewpoint? Is the mutant data presented in the second row.

We apologize for the deficient labelling and therefore have modified the figure adding clearer labels for the viewpoints.

In figure 5b we show two tracks of Virtual 4C using wild-type Hi-C data from S2R+ cells using two different viewpoints (pink rectangles). One viewpoint (top of the Figure 5b; dark blue track) is located at

the B2 boundary, which overlaps a *Notch* enhancer at intron 5. This region interacts not just with the *Notch* locus but also with a gene desert located ~1Mb downstream. The second viewpoint (top of the Figure 5b; purple track) is located at the *CG43689* gene locus just downstream to the 3' end of the gene desert and shows that the region interacts with the near by gene desert and the *Notch* locus. Below the two tracks of Virtual 4C, we included the significant long-range interactions for the B2 region detected by Seqmonk at the fragment level using wild-type Hi-C, which confirms that the B2 boundary establishes significant long-range interactions with a distal genomic region ~1Mb downstream of *Notch* and a zoom into the gene desert displaying several ChIPseq tracks for different APs, histone marks and RNA Pol II.

Updated Figure 5b

4. For the differentially expressed genes (figure 5C), I would like to see the logFC and p-value volcano plot of all genes in the locus, just to give a better representation of the argument the authors try to make.

We have included a volcano plot of all differentially expressed genes located in the mega-base sized domain (from *Notch* to *CG43689*). As you can observe most of the genes located at the domain that are differentially expressed become downregulated after deletion of the B2 boundary (Figure R2-2).

Figure R2-2**Figure R2-2.** Differentially expressed genes within *Notch* megadomain

Figure R2-2. Differentially expressed genes within *Notch* megadomain. This figure is related to Reviewer 2 question 4. Volcano plot of the differentially expressed genes within *Notch* megadomain.

5. Line 457-459. The authors talk about this being conserved in human cells and cite a manuscript in preparation. Since we cannot see that data, I would suggest they either remove the sentence or provide a preprint version of the manuscript so we can verify whether the statement is correct or not.

We agree with the reviewer and have decided to remove the following paragraph from the manuscript:

“Notably, in human cells, CTCF mediates a chromatin loop at the *Notch1* locus, the direct homolog of *Notch* in *Drosophila* (Cerecedo-Castillo et al., in prep), suggesting that CTCF is essential for *Notch* three-dimensional organization across different organisms.”

6. Lines 585-586. In the methods section, the authors mention they computed the correlation between the biological replicates, but they do not provide any parameters the used to compute that. In particular, it is essential to at least know the bin size they used.

We apologize for omitting this information. We calculated the Pearson correlation between our replicates using a bin size of 10 kb. Here is the full parameters used for correlation analysis using HiCexplorer:

```
hicCorrelate -m rep1.h5 rep2.h5 rep3.h5 --method=pearson --log1p --labels rep1 rep2 rep3 --range 5000:200000 --outFileNameHeatmap .pdf --outFileNameScatter.pdf --plotFileFormat pdf
```

7. Figure 5. We had a look at whether the Notch enhancer mega-domain is present in other cells and observed that this is not the case (see attached Figure). Can the authors comment why this locus would show different organization in different cells.

We thank the reviewer for sharing this extra information with us. Indeed, the presence of the mega-domain seems to be a feature of S2R+ cell line (see Figure R2-3a). We speculate that this could be due to several reasons. First, as stated in Answer 2 the B2 boundary of *Notch* is differentially occupied by APs, which could influence the capability of this region to engage in long-range interactions (Figure R2-1c). Second, the binding of APs to the distal end of the mega-domain is also different between cell lines (Figure R2-3c). For example between Kc167 and S2R+ cells we observe differential binding of APs, like CP190 which also correlates with differences in chromatin accessibility at or around the gene desert, in particular at a Long Terminal Repeat (Figure R2-3b). Third, the enrichment of histone modifications at the gene desert is also different between different cell lines (Figure R2-3d), which could influence the cell type specific contacts readily visible upon visual inspection of the HiC matrices (Figure R2-3a).

Figure R2-3

Figure R2-3. The *Notch* megadomain in S2R+ cells

Figure R2-3. The *Notch* megadomain in S2R+ cells. This figure is related to Reviewer 2 question 7. a, Heatmap of Hi-C data binned at 20 kb resolution for different stages of embryonic development and cell lines, showing the presence of the megadomain in S2R+ cells (black square). b, ATAC-seq profile of the megadomain (top) and the gene desert region (bottom) in Kc167 and S2 cells. c, APs binding profile at the gene desert in Kc and S2 cells. d, Histone modifications profile at the gene desert in Kc167, S2 and BG3 cells.

- **Answers to Reviewer 3**

1. To make the conclusion about RNA Pol II binding more generalizable, can the authors examine this relationship more globally?

We appreciate the reviewer's comment. The relationship between the binding of RNA Pol II and domain boundaries has been extensively documented in *Drosophila*, both during early embryogenesis as well as in cell lines (Hou et al., 2012; Hug et al., 2017; Ramírez et al., 2018; Rowley et al., 2017; Rowley et al., 2019; Sexton et al., 2012; Stadler et al., 2017; Ulianov et al., 2016; Zheng et al., 2019).

For example, Hug et al., 2017 showed that the establishment of domain boundaries during early embryonic development correlates with the binding of RNA Pol II. Importantly, ~90% of boundaries detected in this study at 3-4hrs of embryonic development are located at or near an RNA Pol II peak (Hug et al., 2017). An independent study using early embryos (nuclear cycle 14) also corroborates that domain boundaries are occupied by RNA Pol II (Stadler et al., 2017). In agreement with these findings, domain boundaries identified in *Drosophila* cell lines are also occupied by RNA Pol II, with most of them corresponding to promoter elements and enhancers (Cubebñas-Potts et al., 2017; Hou et al., 2012; Ramírez et al., 2018; Sexton et al., 2012; Ulianov et al., 2016). Furthermore, it has been shown that long-range interactions between promoters and enhancers during embryonic development correlate with the binding of RNA Pol II, independent on the transcriptional state of their target genes (Ghavi-Helm et al., 2014), which agrees with our finding regarding the close relationship between regulatory elements that recruit RNA Pol II and the 3D organization of the genome in *Drosophila*.

In agreement with genome-wide observations, our data show that complete inactivation of domain boundaries at the *Notch* locus correlate with loss of RNA Pol II occupancy (see Figure 2a, f; 4b, e). The series of deletions at the 5' end of *Notch* is particularly informative on the relationship between RNA Pol II occupancy and the establishment of a domain boundary, as partial deletion of the boundary results just in discrete topological changes and reduction in RNA Pol II occupancy. In contrast, the major topological effect observed as fusion of adjacent domains (Figure 2a; N5p-Δ343) correlates with near complete loss of RNA Pol II binding (pSer2 and pSer5).

As requested by the reviewer, to further extend our observations on the relationship between RNA Pol II and the presence of domain boundaries, we identified boundaries in S2R+ cells at 1kb resolution in our *in nucleus* Hi-C wild-type dataset and plotted the fold-enrichment over input of RNA Pol II at the identified boundaries. We also complemented our analysis by including the fold-enrichment over input signal for multiple APs mapped in S2/S2R+ cells.

In agreement with previous genome-wide observations, we found that S2R+ cell domain boundaries are occupied by RNA Pol II, DNA binding Architectural Proteins like M1BP and CTCF, and important co-factors for insulator activity like CP190 (Fig. R3-1a) (Ramírez et al., 2018; Rowley et al., 2017; Stadler et al., 2017; Van Bortle et al., 2014; Wang et al., 2018). Regarding the binding of RNA Pol II, we observe that 84% of the domain boundaries (1684/2011) overlap with an RNA Pol II peak (Fig. R3-1b). The presence of RNA Pol II correlates with stronger insulation activity with boundaries overlapping a RNA Pol II peak showing a significantly lower TAD score value than those lacking RNA Pol II binding (*p-value* 9.922e.11; median TAD score at boundaries with RNA Pol II, -0.366 vs -0.292 median TAD score at boundaries without RNA Pol II) (Fig. R3-1c). Domain boundaries also overlap regions of open chromatin like promoters and enhancers according to chromatin states reported by modENCODE for S2 cells (Fig. R3-1d), consistent with observations in other cell lines (Ramírez et al., 2018). We also identified domain boundaries occupied by the APs BEAF-32, M1BP, CTCF and Su(Hw) that also have motif binding sites for those proteins (suggesting direct binding), and evaluated the co-occupancy of APs, CP190 and RNA Pol II. We found that 93% of boundaries bound by RNA Pol II are also occupied

by CP190 (1563/1684). Consistent with recruitment of CP190 by DNA binding APs, we observe that 100% of boundaries co-occupied by RNA Pol II and CP190 are also bound by at least one AP with motif binding site. Furthermore, 78% of co-occupied boundaries by RNA Pol II and CP190 are also bound by M1BP while 66% show binding and motif binding site for at least M1BP and CTCF, overall suggesting that RNA Pol II and APs occupy domain boundaries in *Drosophila* (Fig. R3-1e).

An example of domain boundaries with different border strength and RNA Pol II binding at the genomic region surrounding E2f1 its presented (Fig. R3-1f). In red, we highlight a domain boundary just downstream of E2f1 with a TAD score of -0.81 which is occupied by RNA Pol II and shows clear insulation (see heatmap) while the weakest boundary in the locus has a TAD score of -0.18 and is not occupied by RNA Pol II, suggesting that the presence of RNA Pol II correlates with boundary strength.

Fig. R3-1

Figure R3-1. The RNA Pol II binds to domain boundaries in S2R+ cells

Figure R3-1. The RNA Pol II binds to domain boundaries in S2R+ cells. This figure is related to Reviewer 3 question 1. a, $\log_2(\text{ChIP}/\text{input})$ signal for RNA Pol II, CP190, mod(mdg4) and multiple DNA binding APs at domain boundaries (+/- 5kb) in S2R+ cells. b, Venn diagram of the overlap between domain boundaries in S2R+ cells and RNA Pol II peaks as identified by MACS2. c, Boxplot of the TAD score values of domain boundaries without or with a binding site for RNA Pol II in S2R+ cells; a Wilcoxon-rank sum test was used to identify significant differences in border strength between both types of domain boundaries. d, Distribution of the chromatin states identified by modENCODE in S2 cells at domain boundaries. e, Overlap of domain boundaries and different combinations of APs, CP190 and RNA Pol II. The number above each bar represents the number of domain boundaries overlapping a given combination of proteins. Top right, boxplots of the TAD score values of domain boundaries with RNA Pol II and without it. Below selected boxplots is shown the median TAD score value of selected categories. A Wilcoxon-rank sum test was used to identify significant differences in border strength between both types of domain boundaries. The *p-value* on the plot indicates a significant difference in TAD score values between selected categories. f, Wild-type Hi-C heatmap at 1kb resolution centred in a domain boundary with a high border strength (negative TAD score) and occupied by APs, CP190 and RNA Pol II. Observe that neighbouring boundaries that have a lower border strength (higher negative TAD score) are not occupied by RNA Pol II.

Finally, it is important to point out that detection of boundaries is highly dependent on the resolution of the Hi-C experiments and the software used to call them, which could result in under or overestimation of the number of detected boundaries, which will impact the interpretation of the results.

2. In particular, the RNA Pol II binding in exon 6 of Notch has not been previously noted, seems very unusual, and should be better investigated. To my knowledge this has not been described in vivo and raises the question as to whether S2R+ cells have an unusual regulation of the Notch gene.

- **Is it truly a promoter?**
- **What RNA species would this make?**
- **Are there RNAs produced in both directions or only towards the 3' of Notch?**
- **Would it be predicted to be translated?**
- **If so, what portion of Notch does this correspond to?**

We appreciate the reviewer's observations and agree that the enrichment of RNA Pol II at exon 6 is uncommon and warrant further investigation. Even though we did not edit this particular region and therefore did not investigate the topological and transcriptional function of it, in order to answer the reviewer's concerns, we have extensively search in the literature as well as analyzed a broad spectrum of datasets as to provide evidence on the existence of additional regulatory elements at *Notch* locus.

In the original series of papers describing the sequence and the expression pattern of the *Notch* locus in *Drosophila*, it was reported that three transcripts originate from the *Notch* locus (Artavanis-Tsakonas et al., 1983; Kidd et al., 1983; Kidd et al., 1986). The most abundant one is a 10.4 kb mRNA, that is developmentally regulated and contains all nine exons of *Notch* (Artavanis-Tsakonas et al., 1983; Kidd et al., 1983; Kidd et al., 1986) while the other two transcripts of ~ 9 and 10 kb are detected just in adult females, unfertilized eggs and early embryos (Kidd et al., 1983; Kidd et al., 1986) (see Figure R3-2a,b). According to Kidd et al., 1986, the 9 kb transcript lack the first five exons of *Notch*, which would result in a protein without the hydrophobic signal localization peptide and the first six EGF-like repeats of the extracellular domain of *Notch* (Kidd et al., 1986) (see Figure R3-2c). If such isoform is translated and functional is unknown, however, the presence of transcript isoforms of *Notch* at specific stages of development, suggest either alternative splicing or the existence of stage specific transcriptional regulation. In fact, in Kidd et al., 1986 it was stated in reference to the 9 kb transcript of *Notch*: **"whether this transcript is initiated upstream of exon 6 or is produced by alternative splicing is unknown"** (Kidd et al., 1986).

Given the early observations that suggest the existence of an alternative isoform of *Notch* in adult females and unfertilized eggs, we turned to publicly available ChIP-chip datasets of virgin female ovaries from modENCODE to look for chromatin features that could suggest the existence of an alternative transcriptional unit around exon 6 of *Notch*. We observe an enrichment for the “active” histone post-translational modification H3K27ac at exon 6, similar to what we observe in the S2R+ cell line (see Figure R3-2d) while we also detected a progressive enrichment of “active” histone post-translational modifications in the region comprising exon 6-9 during embryonic development (Figure R3-2e) in a pattern similar to what we observe for the same region in S2R+ cells (see Figure 1a and Figure R1-1a). Also based on modENCODE data, we detected a dynamic repositioning of the RNA Pol II at the 5' end of *Notch* in the region comprising exon 6-9 which seems to be independent on the chromatin accessibility at the locus (see Figure R3-2f). Furthermore, we observe a differential exon usage for *Notch* during fly development (<http://flybase.org/cgi-bin/rnaseqmapper.pl>). As observed in Figure R3-2g, during early embryonic development the transcript levels of each of the 9 exons of *Notch* is similar. However, from embryos at 14 hrs of development and later, as well as in certain stages of post-embryonic development, we observe that exons 6, 7 and 8 show higher levels of transcription than the rest of the exons (embryos 4-6 hrs after egg laying vs L3 larvae gut and white prepupae). Interestingly, in S2R+ cells we detected that exons 6-9 are also transcribed at higher level than the first five exons as evaluated by RNA-seq (see Figure 5d and Figure R1-1a). Hence, the production of alternative transcripts at *Notch* has been reported before *in vivo*, however the function of these transcripts remains unknown.

Regarding the specific case of *Notch* in the S2R+ cell line, we detect that RNA Pol II binds to at least three regions at *Notch*: the 5' end, an enhancer at intron 5 and the start of exon 6 (Figure 1b and Figure R1-1a). The presence of multiple sites of RNA Pol II enrichment at the *Notch* locus suggests the existence of additional regulatory elements in addition to the promoter of *Notch*. As stated in Answer 1 for Reviewer 1 we observe that the promoter of *Notch*, the enhancer at intron 5 and the region occupied by RNA Pol II at the exon 6 are transcribed in both directions as evidenced by nascent RNA data derived from GRO-seq and START-seq experiments in S2 cells (Figure R1-1a). The presence of bidirectional transcription has been recently proposed as a proxy to identify regulatory elements independent on additional chromatin features (Andersson and Sandelin, 2019). Also, we detected by *in nucleus* Hi-C, that exon 6 to 9 are located within a contact domain (Domain 2) in S2R+ cells and other cell types including specific stages of the fly life cycle *in vivo* (see Figure R2-1 and Supplementary Figure 2). The higher expression of exons located in domain 2 and their sustained expression despite removal of the B1 boundary of *Notch*, suggest that additional regulatory elements within the locus can drive transcription of *Notch* exons located in Domain 2 (see Figure 3b, Figure R1-1a). In support of this, deletion of the enhancer element located at intron 5 results not just in a gain of interactions between the Domain 1 and 2 of *Notch* but also in a marked decrease of transcription of exons within Domain 2 and a major reduction in RNA Pol II occupancy at exon 6 (See Figure 4). Interestingly, it has been reported that enhancers can act as promoters in *Drosophila* (Mikhaylichenko et al., 2018). Then, these evidences suggest that in S2R+ cells there are regulatory elements that can recruit RNA Pol II and produce RNA species at the *Notch* locus. Furthermore, our data show that the RNA Pol II recruitment at exon 6 is in part dependent on the presence of the enhancer at intron 5 as RNA Pol II enrichment is dramatically reduced upon enhancer deletion (Figure 4e).

Despite the above evidences from the literature, since we did not functionally tested the exon 6 region for promoter activity nor performed genomic edition at this genomic region, and given our results regarding non-autonomous recruitment of RNA Pol II at exon 6 upon enhancer deletion, we have substituted in the manuscript the term “Putative promoter or intergenic promoter” just for exon 6. Also, given our observations on minor transcriptional effects at exons within Domain 2 upon disruption of the 5' boundary of *Notch* we have added a plot of the qChIP results for H3K27ac and RNA Pol II at the enhancer at intron 5 (Boundary 2) (Fig.2 f) that shows that RNA Pol II enrichment at this regulatory

element is maintained in the 5' CRISPR mutants and therefore suggests that this regulatory element could be important to maintain transcription of exons within Domain 2 in S2R+ cells.

Fig. R3-2

Figure R3-2. The transcriptional and chromatin landscape of the *Notch* locus in *Drosophila melanogaster*

Figure R3-2. The transcriptional and chromatin landscape of the *Notch* locus in *Drosophila melanogaster*. This figure is related to Reviewer 3 question 2. a, Northern Blot using RNA derived from wild-type embryos, larvae, pupa and adults, to characterize *Notch* transcriptional dynamic during development (modified from (Kidd et al., 1983)). b, Northern Blot using RNA derived from wild-type embryos, larvae, pupa and adults, to characterize *Notch* transcriptional dynamic during development (modified from (Artavanis-Tsakonas et al., 1983)). c, Structure of the *Notch* locus and the positions of identified protein domains. Shown are the exons of *Notch* and regions that would not be included in the shorter isoform of *Notch* (modified from (Kidd et al., 1986)). d, H3K27ac enrichment profile and CAGE signal at the *Notch* locus in virgin female ovaries (data derived from modENCODE). e, Enrichment profile of the histone modifications: H3K4me3, H3K4me1, H3K27ac, H3K27me3, H3K9me3 and RNA-seq data at the *Notch* locus during *D. melanogaster* embryonic development (data derived from modENCODE). Domain 1 and 2 correspond to the regions identified as topological domains in S2R+ cells (this study). f, Chromatin accessibility (DNA-seq) and enrichment profile of RNA-Pol II at *Notch* locus during embryonic development (data derived from modENCODE). Domain 1 and 2 correspond to the regions identified as topological domains in S2R+ cells (this study). g, Screenshot of the RPKM values of *Notch* exons during *D. melanogaster* development as obtained from Flybase (data derived from modENCODE). On top is shown a cartoon of the *Notch* locus and each of its exons.

3. The transposable elements in this region are ignored in the manuscript. I feel it is important to include these on the genome maps, annotated from their DNaseq data. Those of S2R+ cells may differ widely from the reference genome. Given the recent suggestion in mammalian cells that retrotransposons may be important boundary elements, these sequences should be positioned on the coordinates. Where are TEs positioned in reference to the numerous promoter, enhancer and AP binding regions?

We appreciate the reviewer's suggestion and agree that the role of transposable elements in genome organization should be addressed thoughtfully in *Drosophila* as recent reports suggest a contribution of these elements in genome organization in mammals (Kaaij et al., 2019; Kruse et al., 2019; Zhang et al., 2019). In this regard, we believe it is a very interesting venue of research and would like to pursue it in the near future, however, we think performing a genome wide annotation of TEs is out of the scope of the present work.

However, in order to try to answer the reviewer's concerns about the possible presence of transposable elements at the *Notch* locus, in particular at regulatory elements functionally analysed in the present study, we have performed additional analysis (see Figure R3-3). First we looked for annotated repeat sequences by RepeatMaker in the *D. melanogaster* reference genome (dm6) at the *Notch* locus (Figure R3-3a). Based on annotations by RepeatMasker in the dm6 reference genome we do not observe any sequence belonging to transposable elements of the LINE, SINE or LTR families of retrotransposable elements, which have been reported to impact 3D genome organization in mammals (Kruse et al., 2019; Zhang et al., 2019). However, we do observe the presence of simple repeats as detected by RepeatMasker (<http://www.repeatmasker.org/faq.html>) and defined as duplications of simple sets of DNA bases (typically 1-5bp) such as A, CA, CGG. The presence of simple repeats is detected along the *Notch* locus and they are also detected in regions surrounding *Notch* (Figure R3-3a).

The 5' intergenic region of *Notch* contains a 47-bp direct repeat originally described by Ramos et al., 1989. We confirmed the presence of such sequence in the reference genome by using the Simple Tandem Repeats track from UCSC (Figure R3-3b). This sequence is eliminated as part of the CRISPR 5' mutants with exception of the *N5p-Δ102*. Importantly there is no binding site for any reported AP detected at this sequence by JASPAR (Figure R3-3b). These suggest that the topological and transcriptional effects observed in the CRISPR mutants are not due to removal of a binding motif for APs at this tandem repeat. Also, we do not observe tandem repeats at the enhancer in intron 5 or at exon 6 (Figure R3-3c). Importantly, many other tandem repeats are detected along the *Notch* locus or in neighboring loci and do not overlap with domain boundaries or binding of APs (Figure R3-3a).

As the above analysis was performed using the reference dm6 genome, we reasoned that the presence of an S2R+ specific transposable element at *Notch* domain boundaries should be readily detected by PCR as a band of higher than the expected size using primers flanking each boundary and genomic DNA from wild-type S2R+ cells. In this regard, primers used for genotyping CRISPR mutants at the B1 boundary always resulted in the expected amplicon size for wild-type alleles according to the reference genome (Figure 1b, B1 boundary: ~1500 bp). An additional primer pair flanking the genomic region spanning the B1 boundary, also result in the expected wild-type size (Figure R3-3d; 970 bp). For the B2 boundary at intron 5 we observe the expected wild-type size when using primers for genotyping (Figure R3-3d; P1-3372 bp) and two additional set of pairs of primers flanking the accessibility sites corresponding to the regulatory element removed in the CRISPR- Δ enh mutant (Figure R3-3d; P2-1842 bp and P3-1247 bp). Moreover, all CRISPR clones were sequenced retrieving the expected nucleotides according to the dm6 genome, which suggest that the regulatory elements analysed in our study do not harbour cell-type specific insertions of transposable elements (Supplementary Figure 3).

Fig. R3-3

a. Notch locus

**b. Boundary B1
5' end Notch**

**c. Boundary B2
Enhancer - Intron 5**

d.

5' Notch
Fwd:TTTGTA AAAACATTTGAGCGATCT
Rev:TCTGTTTCAAATCGGCAGTG

P2
Fwd:CAGATTCAAAGTGGCGTCA
Rev:AGTTCCTGGCCCTCGAAACCT

P1
Fwd:GGTTACCCACGGAAATTCGAG
Rev:ACATCGTCTGGCAGAATCT

P3
Fwd:TTGAAGCCCTCTTTGTGG
Rev:CACTCCCCTGCTGACTAAG

Figure R3-3. Transposable elements and repeat sequences at the *Notch* locus in *Drosophila melanogaster*

Figure R3-3. Transposable elements and repeat sequences at the *Notch* locus in *Drosophila melanogaster*. This figure is related to Reviewer 3 question 3. a, Repeat elements, simple tandem repeats and microsatellites at the *Notch* locus (Data obtained from UCSC genome browser for dm6). b, Same as a but at the 5' region of *Notch* (Boundary B1). Also is shown the TBBS prediction by JASPAR 2020. Highlighted in red and flanked by scissors is the full region covered by our CRISPR mutants. c, Same as a but at the region at the B2 boundary of *Notch* (intron 5 – exon 6). Highlighted in red and flanked by scissors is the full region covered by our CRISPR mutants. d, Top, ATAC-seq profile in S2 cells at the regions shown in b and c. 1% Agarose gels showing PCR products derived from wild-type S2R+ genomic DNA using primers spanning the B1 and B2 Boundaries analysed in this study. Sequences of the primers are shown below agarose gels.

Finally, we do not detect gaps in our sequencing results from Hi-C (see Figure 1a) nor insertions are detected at this region in S2 cells based on DNA-seq (see (Ulianov et al., 2016), Supplementary Table 4).

Hence, our data suggest that transposable elements are not present at the Domain Boundaries of the *Notch* locus in the reference genome, nor in the S2R+ cell. We also reasoned that as the topological structures evaluated at the *Notch* locus are not specific of S2R+ cells, as they are present in embryos and in other cell lines (see Figure R2-1), the direct contribution of TEs to the *Notch* domain 3D structure particularly at S2R+ cells seems unlikely. However and as pointed out by the reviewer, we do think this could be happening in other regions in the genome and as stated, consider this a very interesting venue of research for future studies.

4. There seem to be differences in Notch TADs and Pol II binding in Kc cells from Ramirez F, Nat Comm. Can the authors comment on these differences?

We agree with the reviewer. We believe that differences in RNA Pol II binding between both cell lines could be related in part due to differences in 3D genome organization, recruitment of APs at domain boundaries and differential transcription as Kc167 cells are of female origin in contrast to S2/S2R+ which are of male origin (find the full answer in Answer 1 to Reviewer 2 and Figure R2-1).

5. On p7 they state that the establishment of boundaries at Notch strongly correlates with the progressive gain of chromatin accessibility and the binding of Zelda and RNA-Pol II I do not see the evidence for Pol II at the "putative internal promoter" in these data. Why is this?

We agree with the reviewer and for clarity changed the phrase:

In Line 166: “We found that the establishment of boundaries at *Notch* strongly correlates with the progressive gain of chromatin accessibility and the binding of Zelda and RNA-Pol II (Supplementary Fig. 2b)”

for:

“We found that the establishment of boundaries at Notch strongly correlates with the progressive gain of chromatin accessibility and the binding of Zelda up to nc14 (Supplementary Fig. 2b)”

We propose that the lack of RNA Pol II at the exon 6 at this stage could be related with the transcriptional status of *Notch* depending on the stage of embryonic development. In supplementary Figure 2b we show the progressive gain of chromatin accessibility at both B1 and B2 boundaries during early embryonic development. In particular it is observed that by nc14, and concomitant with zygotic genome activation, the B1 boundary of Notch is occupied by Zelda and RNA Pol II while the B2

boundary shows a modest enrichment of Zelda. The differences observed with S2R+ could be related with transcription of the *Notch* locus as the maximum transcriptional activity is detected later in embryonic development (4-8 hrs) with coincides with a clear definition of the two domains of *Notch* (See Supplementary Figure 2a). We also observe that RNA Pol II becomes enriched at exon 6 at 8-10hrs of embryonic development (see Figure R3-2 f) and that during embryonic development the *Notch* locus shows a dynamic repositioning of active histone modifications (Figure R3-2 e).

6. It is unclear in the methods if the CRISPR cell lines are from single cell clones or not. Are they working with mixed populations of cells or not? Additionally, I believe that S2R+ cells are male, but tetraploid. Therefore there are 2 X chromosomes- are they both mutated in the same way?

We apologize for the misunderstanding. As stated in the Methods section:

Line 567:

“Single clones were expanded and an aliquot of cells was 567 used for DNA extraction by phenol-chloroform and PCR genotyping using specific primers spanning the desired deletions (see Table 2). Homozygous clones were expanded and used for subsequent experiments. Mutant clones were further characterized by Sanger Sequencing using primers for genotyping. Electropherograms of breakpoints for all CRISPR mutants generated in this study are shown in Supplementary Fig. 3.”

All our CRISPR results are derived from single clones isolated by serial dilutions from pools of CRISPR mutant cells. We validated the presence of just the mutant band expected from each mutant by PCR in conjunction with the absence of the wild-type band (see for example Figure 1c). We also determined the identity of the CRISPR mutant by Sanger Sequencing of the mutant band obtained by PCR.

7. Many of the Figures need to be enlarged and with better resolution: Sup 1c and sup 2b for example. Also a lot of important data are in Sup Fig 5A, which is difficult to see since the resolution is bad. It would help to separately letter each subfigure so it can be referenced separately in the text.

We agree with the reviewer. We have made the appropriate changes.

8. It might be helpful to plot some of the data as log fold changes (Figure 1B for example), which would make the significant changes in CTCF of K27me3 more visible.

We appreciate the reviewer's comment. We generated plots using the log fold changes (see examples in Figure R3-4) and for visualization purposes we decided to keep the original scale.

Figure R3-4

Figure R3-4 Examples of log2 Fold Enrichment ChIP-qPCR experiments

References

- Andersson, R. and Sandelin, A.** (2019). Determinants of enhancer and promoter activities of regulatory elements. *Nat. Rev. Genet.* 1–17.
- Artavanis-Tsakonas, S., Muskavitch, M. A. and Yedvobnick, B.** (1983). Molecular cloning of Notch, a locus affecting neurogenesis in *Drosophila melanogaster*. *Proc. Natl. Acad. Sci. U. S. A.* **80**, 1977–1981.
- Chathoth, K. T. and Zabet, N. R.** (2019). Chromatin architecture reorganization during neuronal cell differentiation in *Drosophila* genome. *Genome Res.* **29**, 613–625.
- Cubeñas-Potts, C., Rowley, M. J., Lyu, X., Li, G., Lei, E. P. and Corces, V. G.** (2017). Different enhancer classes in *Drosophila* bind distinct architectural proteins and mediate unique chromatin interactions and 3D architecture. *Nucleic Acids Res.* **45**, 1714–1730.
- Ghavi-Helm, Y., Klein, F. A., Pakozdi, T., Ciglar, L., Noordermeer, D., Huber, W. and Furlong, E. E. M.** (2014). Enhancer loops appear stable during development and are associated with paused polymerase. *Nature* **512**, 96–100.
- Gutierrez-Perez, I., Rowley, M. J., Lyu, X., Valadez-Graham, V., Vallejo, D. M., Ballesta-Illan, E., Lopez-Atalaya, J. P., Kremsky, I., Caparros, E., Corces, V. G., et al.** (2019). Ecdysone-Induced 3D Chromatin Reorganization Involves Active Enhancers Bound by Pipsqueak and Polycomb. *Cell Rep.* **28**, 2715-2727.e5.
- Hou, C., Li, L., Qin, Z. S. and Corces, V. G.** (2012). Gene density, transcription, and insulators contribute to the partition of the *Drosophila* genome into physical domains. *Mol. Cell* **48**, 471–484.
- Hug, C. B., Grimaldi, A. G., Kruse, K. and Vaquerizas, J. M.** (2017). Chromatin Architecture Emerges during Zygotic Genome Activation Independent of Transcription. *Cell* **169**, 216-228.e19.
- Kaaij, L. J. T., Mohn, F., van der Weide, R. H., de Wit, E. and Bühler, M.** (2019). The ChAHP Complex Counteracts Chromatin Looping at CTCF Sites that Emerged from SINE Expansions in Mouse. *Cell* **178**, 1437-1451.e14.
- Kidd, S., Lockett, T. J. and Young, M. W.** (1983). The Notch locus of *Drosophila melanogaster*. *Cell* **34**, 421–433.
- Kidd, S., Kelley, M. R. and Young, M. W.** (1986). Sequence of the notch locus of *Drosophila melanogaster*: relationship of the encoded protein to mammalian clotting and growth factors. *Mol. Cell. Biol.* **6**, 3094–3108.
- Kruse, K., Díaz, N., Enriquez-Gasca, R., Gaume, X., Torres-Padilla, M.-E. and Vaquerizas, J. M.** (2019). Transposable elements drive reorganisation of 3D chromatin during early embryogenesis. *bioRxiv* 523712.
- Lee, H., McManus, C. J., Cho, D.-Y., Eaton, M., Renda, F., Somma, M. P., Cherbas, L., May, G., Powell, S., Zhang, D., et al.** (2014). DNA copy number evolution in *Drosophila* cell lines. *Genome Biol.* **15**, R70.

- Li, L., Lyu, X., Hou, C., Takenaka, N., Nguyen, H. Q., Ong, C.-T., Cubeñas-Potts, C., Hu, M., Lei, E. P., Bosco, G., et al. (2015). Widespread Rearrangement of 3D Chromatin Organization Underlies Polycomb-Mediated Stress-Induced Silencing. *Mol. Cell* **58**, 216–231.
- Liang, J., Lacroix, L., Gamot, A., Cuddapah, S., Queille, S., Lhoumaud, P., Lepetit, P., Martin, P. G. P., Vogelmann, J., Court, F., et al. (2014). Chromatin Immunoprecipitation Indirect Peaks Highlight Long-Range Interactions of Insulator Proteins and Pol II Pausing. *Mol. Cell* **53**, 672–681.
- Lucchesi, J. C. and Kuroda, M. I. (2015). Dosage Compensation in *Drosophila*. *Cold Spring Harb. Perspect. Biol.* **7**.
- Mikhaylichenko, O., Bondarenko, V., Harnett, D., Schor, I. E., Males, M., Viales, R. R. and Furlong, E. E. M. (2018). The degree of enhancer or promoter activity is reflected by the levels and directionality of eRNA transcription. *Genes Dev.* **32**, 42–57.
- Nagano, T., Lubling, Y., Stevens, T. J., Schoenfelder, S., Yaffe, E., Dean, W., Laue, E. D., Tanay, A. and Fraser, P. (2013). Single-cell Hi-C reveals cell-to-cell variability in chromosome structure. *Nature* **502**, 59–64.
- Nagano, T., Várnai, C., Schoenfelder, S., Javierre, B.-M., Wingett, S. W. and Fraser, P. (2015). Comparison of Hi-C results using in-solution versus in-nucleus ligation. *Genome Biol.* **16**, 175.
- Ong, C.-T., Van Bortle, K., Ramos, E. and Corces, V. G. (2013). Poly(ADP-ribosyl)ation Regulates Insulator Function and Intrachromosomal Interactions in *Drosophila*. *Cell* **155**, 148–159.
- Ramírez, F., Bhardwaj, V., Arrigoni, L., Lam, K. C., Grüning, B. A., Villaveces, J., Habermann, B., Akhtar, A. and Manke, T. (2018). High-resolution TADs reveal DNA sequences underlying genome organization in flies. *Nat. Commun.* **9**, 189.
- Rao, S. S. P., Huntley, M. H., Durand, N. C., Stamenova, E. K., Bochkov, I. D., Robinson, J. T., Sanborn, A. L., Machol, I., Omer, A. D., Lander, E. S., et al. (2014). A 3D Map of the Human Genome at Kilobase Resolution Reveals Principles of Chromatin Looping. *Cell* **159**, 1665–1680.
- Rowley, M. J., Nichols, M. H., Lyu, X., Ando-Kuri, M., Rivera, I. S. M., Hermetz, K., Wang, P., Ruan, Y. and Corces, V. G. (2017). Evolutionarily Conserved Principles Predict 3D Chromatin Organization. *Mol. Cell.*
- Rowley, M. J., Lyu, X., Rana, V., Ando-Kuri, M., Karns, R., Bosco, G. and Corces, V. G. (2019). Condensin II Counteracts Cohesin and RNA Polymerase II in the Establishment of 3D Chromatin Organization. *Cell Rep.* **26**, 2890-2903.e3.
- Sexton, T., Yaffe, E., Kenigsberg, E., Bantignies, F., Leblanc, B., Hoichman, M., Parrinello, H., Tanay, A. and Cavalli, G. (2012). Three-dimensional folding and functional organization principles of the *Drosophila* genome. *Cell* **148**, 458–472.
- Stadler, M. R., Haines, J. E. and Eisen, M. B. (2017). Convergence of topological domain boundaries, insulators, and polytene interbands revealed by high-resolution mapping of chromatin contacts in the early *Drosophila melanogaster* embryo. *eLife* **6**, e29550.
- Ulianov, S. V., Khrameeva, E. E., Gavrilov, A. A., Flyamer, I. M., Kos, P., Mikhaleva, E. A., Penin, A. A., Logacheva, M. D., Imakaev, M. V., Chertovich, A., et al. (2016). Active chromatin and transcription play a key role in chromosome partitioning into topologically associating domains. *Genome Res.* **26**, 70–84.

- Van Bortle, K., Nichols, M. H., Li, L., Ong, C.-T., Takenaka, N., Qin, Z. S. and Corces, V. G.** (2014). Insulator function and topological domain border strength scale with architectural protein occupancy. *Genome Biol.* **15**, R82.
- Vogelmann, J., Gall, A. L., Dejardin, S., Allemand, F., Gamot, A., Labesse, G., Cuvier, O., Nègre, N., Cohen-Gonsaud, M., Margeat, E., et al.** (2014). Chromatin Insulator Factors Involved in Long-Range DNA Interactions and Their Role in the Folding of the *Drosophila* Genome. *PLOS Genet.* **10**, e1004544.
- Wang, Q., Sun, Q., Czajkowsky, D. M. and Shao, Z.** (2018). Sub-kb Hi-C in *D. melanogaster* reveals conserved characteristics of TADs between insect and mammalian cells. *Nat. Commun.* **9**, 188.
- Wood, A. M., Van Bortle, K., Ramos, E., Takenaka, N., Rohrbaugh, M., Jones, B. C., Jones, K. C. and Corces, V. G.** (2011). Regulation of Chromatin Organization and Inducible Gene Expression by a *Drosophila* Insulator. *Mol. Cell* **44**, 29–38.
- Zhang, Y., Li, T., Preissl, S., Amaral, M. L., Grinstein, J. D., Farah, E. N., Destici, E., Qiu, Y., Hu, R., Lee, A. Y., et al.** (2019). Transcriptionally active HERV-H retrotransposons demarcate topologically associating domains in human pluripotent stem cells. *Nat. Genet.* **51**, 1380–1388.
- Zheng, M., Tian, S. Z., Capurso, D., Kim, M., Maurya, R., Lee, B., Piecuch, E., Gong, L., Zhu, J. J., Li, Z., et al.** (2019). Multiplex chromatin interactions with single-molecule precision. *Nature* **566**, 558–562.

REVIEWERS' COMMENTS:

Reviewer #1 (Remarks to the Author):

The authors have done an excellent job at addressing all my concerns as well as the concerns of the other reviewers. The manuscript represents an important contribution to the field and is appropriate for publication in Nat Comm

Reviewer #2 (Remarks to the Author):

Overall, I am happy with the additional analysis performed by the authors to address the points raised by me. I think the results they found are very interesting and should be included in the manuscript (at least as supplementary Figures).

1. (point 1) Figure R2-1 and the corresponding analysis: It is key to show that there is strong link between these domains and H4K16ac and dosage compensation in *Drosophila*.
2. (point 4) Figure R2-2 shows clearly the differential expression following a standard approach. I think it should be included in the manuscript as it makes the claims clearer to follow.
3. (point 6) The authors explained the bins used for correlation analysis, but I could not see this added in the main manuscript. I think it should be added
4. (point 7) Figure R2-3 compares the presence of these loci in different cells/stages and I think it is important to add this to the manuscript as it provides stronger evidence for cell specific roles of 3D chromatin organisation. This is key if the authors want to link 3D chromatin organisation, enhancers and gene expression as conferring cell specificity.

Reviewer #3 (Remarks to the Author):

I appreciate the authors' detailed rebuttal and additional figures for reviewers. They have addressed a vast majority of my major points.

I do feel, however, that they could be clearer or more precise in the manuscript regarding the nature of the cells they are examining:

They did not address my question of their cell lines being tetraploid and whether each X chromosome is mutated in the same way (point 6). This should be better described in the methods. In addition, the point raised by Reviewer 2 and the discussion with the authors about potential differences in male vs female cells is important. I feel that they do not mention this in the manuscript. Because male/female differences might make a large difference, this fact should be put in the discussion and (or minimally) in the methods. This will allow readers to understand this

immediately. It does not detract from the manuscript, but may end up being an important difference between data sets of different cell lines or from in vivo tissues.

There are some minor English points that would be good to correct. Here are a few:

line 58- "precluding" instead of "preclude"

line 71: "decrease in TAD boundaries insulation" I would say "decrease in TAD boundary insulation" or "decrease in insulation of TAD boundaries"

line 72: "On the other hand, depletion of BEAF-32, AP enriched at TAD boundaries, result in minor changes in genome organization" should be "On the other hand, depletion of BEAF-32, an AP enriched at TAD boundaries, results in minor changes in genome organization"

line 111: "Quality controls of all replicates are of highest standards," I would avoid subject statements like this in the results and simply state that "a minimum of 89% valid pairs per replicate was achieved"

Reviewer #1 (Remarks to the Author):

The authors have done an excellent job at addressing all my concerns as well as the concerns of the other reviewers. The manuscript represents an important contribution to the field and is appropriate for publication in Nat Comm.

We appreciate the Reviewer #1 final comments and we are glad we have addressed all her/his concerns. Importantly, we also would like to acknowledge her/his commitment in reviewing our work and her/his comments that improved the current version of our manuscript.

Reviewer #2 (Remarks to the Author):

Overall, I am happy with the additional analysis performed by the authors to address the points raised by me. I think the results they found are very interesting and should be included in the manuscript (at least as supplementary Figures).

We appreciate the Reviewer #2 final comments and we are glad we have addressed all her/his concerns. Importantly, we also would like to acknowledge her/his commitment in reviewing our work and her/his comments that improved the current version of our manuscript. Following the Reviewer # 2 suggestions we will include the Figures R2-1, 2, and 3 as Supplementary Figures and answer specific concerns below.

Specific answers to Reviewers #2:

1. (point 1) Figure R2-1 and the corresponding analysis: It is key to show that there is strong link between their domains and H4K16ac and dosage compensation in Drosophila. 2.

We have included Figure R2-1 as **Supplementary Figure 3**. With the title:

The 3D organization of the Notch locus in D. melanogaster is different between female and male cells and topological differences correlate with the enrichment of H4K16ac and the binding of Architectural Proteins.

Supplementary Figure 3 does not contain the plots presented in panel **a** from Figure R2-1 as they have also been included in Supplementary Figure 2 of our manuscript. The accession numbers for all the datasets presented in Figure R2-1 have also been included in the Methods section. We have also included in the manuscript the analysis of the Supplementary Figure 3 in the Results section “**Notch is organized into two topological domains in S2R+ cells**” as follows:

In *Drosophila*, dosage compensation occurs at active loci of the male X chromosome due to the recruitment of the MSL complex which results in increased transcription and the deposition of the histone post-translational modification H4K16ac into the gene bodies of dosage compensated loci^{36,37}. Recent observations suggest the existence of sex-specific differences in genome organization between the X chromosomes of female and male cells, which correlate with differences in the binding of APs at domain boundaries and with the differential enrichment of H4K16ac at dosage compensated loci^{38,39}. The *Notch* locus is located in the X chromosome and shows differential enrichment of H4K16ac as well as different expression level between male and female suggesting is dosage compensated (Supplementary Fig. 3a). To investigate differences in genome organization at the *Notch* locus between female and male cells we re-analysed Hi-C datasets derived from the female embryonic cell line Kc167^{22,40} and the male CNS-L3 cell line BG3⁴¹ and compared *Notch* 3D organization with the one observed in the male embryonic cell line S2R+. We observe two domains spanning the *Notch* locus in the male derived cell lines (S2R+ and BG3) in contrast with a single domain organization of *Notch* in the female derived cell line Kc167 (Supplementary Fig. 3a). To relate the observed topological differences at *Notch* with the presence of the histone post-translational modification H4K16ac and the binding of APs we obtained processed signal files from modENCODE for H4K16ac and re-analysed publicly available ChIP-seq data sets of APs for the Kc167 cell line^{22,40,42}. We observe a clear difference in the enrichment of H4K16ac between the female cell line (Kc167) and the male cell lines (S2R+ and BG3), with H4K16ac being highly enriched at the *Notch* region encompassing the Domain 2 detected in male cells (exon 6 – exon 9) while the histone post-translational modification H3K27me3 is enriched at the same region in female cells and overall depleted in male cells (Supplementary Fig. 3a). Furthermore, the two domain organization of *Notch* and the enrichment of H4K16ac at Domain2 in male cells correlate with higher expression levels of *Notch* when compared to a female derived cell line (Supplementary Fig. 3a). We also observe differences in APs occupancy at the *Notch* locus between female and male cell types which correlates with the presence of domain boundaries along the locus (Supplementary Fig. 3b,c). In particular, while the B1 at the 5' end of *Notch* is observed in both female and male derived cell lines and this correlates with the binding of multiple APs, CP190 and RNA Pol II, the genomic region encompassing the B2 boundary in male cells shows an overall reduction in the binding of APs in the female derived Kc167 cell line (Supplementary Fig. 3c). Importantly, the observed difference in APs binding at the B2 boundary genomic region in Kc167 cells is not due to differential chromatin accessibility since Kc167 and S2 cells show a remarkable similar ATAC-seq profile at the *Notch* locus (Supplementary Fig. 3b,c)⁴³. The dynamic organization of this locus contrast with early observations suggesting a mostly invariant organization of the genome in *Drosophila* and supports recent observations that boundaries are dynamic between cell types and that these variability correlates with the binding of APs⁴¹

Together, these results indicate that the *Notch* locus is organized into two topological domains that isolate the gene from neighboring TADs in S2R+ cells. *Notch* domain boundaries are enriched for

active histone marks and occupied by RNA Pol II and for a variable number of Architectural Proteins. During embryonic development, domain boundaries are detected before transcription of the locus and strongly correlate with the progressive acquisition of chromatin accessibility and RNA Pol II binding. Also, the *Notch* locus shows sex-specific topological organization which correlates with differences in transcription, the enrichment of histone modifications like H4K16ac and the binding of APs.

Also, given the new **Supplementary Figure 3** we have changed the numbering of the following Supplementary Figures without altering their content.

Name in previous version of manuscript	Name in present version of manuscript
Supplementary Figure 3	Supplementary Figure 4
Supplementary Figure 4	Supplementary Figure 5
Supplementary Figure 5	Supplementary Figure 6

2. (point 4) Figure R2-2 shows clearly the differential expression following a standard approach. I think it should be included in the manuscript as it makes the claims clearer to follow.

Please see 4. of this letter for a detailed explanation of the new Supplementary Figure related to R2-2.

3. (point 6) The authors explained the bins used for correlation analysis, but I could not see this added in the main manuscript. I think it should be added.

We apologize for missing to add the details of our correlation analysis in the manuscript as provided in the Answer to Reviewers file. We have added the missing information in the Methods section:

Correlation plots between all Hi-C replicates and counts vs distance plots were generated using HiCExplorer¹⁸ **and matrices with a bin size of 10 kb.**

4. (point 7) Figure R2-3 compares the presence of this loci in different cells/stages and I think it is important to add this to the manuscript as it provides stronger evidence for cell specific roles of 3D chromatin organisation. This is key if the authors want to link 3D chromatin organisation, enhancers and gene expression as conferring cell specificity.

We appreciate the reviewer recommendation. Since Figures R2-2 and R2-3 are directly related with the *Notch* megadomain and to ensure a logic presentation of our data we have created three new Supplementary Figures termed **Supplementary Figure 7,8,9.**

Name in previous version of manuscript	Name in present version of manuscript
Supplementary Figure 6	Supplementary Figure 7. Deletion of the B2 boundary of Notch results in local and long-range topological defects.
New	Supplementary Figure 8. The Notch megadomain is unique to S2R+ cells and its anchors show differential chromatin accessibility, binding of Architectural Proteins and histone marks between different cell types.
New	Supplementary Figure 9. Deletion of the B2 boundary in S2R+ cells eliminates the Notch megadomain and affects transcription of the

genes located within the megadomain.

Supplementary Figure 7. In the previous version of our manuscript, the Supplementary Figure 6 presented plots related to the local effects on *Notch* topology after deletion of B2 boundary (panel a, b, c, d) as well as the effects in the *Notch* megadomain (panel e, f, and g). Given the Figures R2-2 and R2-3 are directly related to the *Notch* megadomain we have decided to present in Supplementary Figure 7 just information about local effects upon B2 deletion (panels a, b, c, d from the Supplementary Figure 6 presented in the previous version of our manuscript).

Supplementary Figure 8. This figure contains all panels presented as Figure R2-3 in our rebuttal letter. Since data presented here concerns the presence of the *Notch* megadomain in other cell types as well as the chromatin characterization of the megadomain anchors we felt it was better to present this information as a single figure which will be referenced in the main text before discussion of the topological and transcriptional effects observed upon loss of the megadomain.

Supplementary Figure 9. This figure contains all additional plots related to the topological and transcriptional effects observed upon loss of the megadomain as well as the volcano plot presented as Figure R2-2 in our rebuttal letter.

Reviewer #3 (Remarks to the Author):

I appreciate the authors' detailed rebuttal and additional figures for reviewers. They have addressed a vast majority of my major points.

I do feel, however, that they could be clearer or more precise in the manuscript regarding the nature of the cells they are examining:

They did not address my question of their cell lines being tetraploid and whether each X chromosome is mutated in the same way (point 6). This should be better described in the methods. In addition, the point raised by Reviewer 2 and the discussion with the authors about potential differences in male vs female cells is important. I feel that they do not mention this in the manuscript. Because male/females differences might make a large difference, this fact should be put in the discussion and (or minimally) in the methods. This will allow readers to understand this immediately. It does not detract from the manuscript, but may end up being an important difference between data sets of different cell lines or from in vivo tissues.

We appreciate the Reviewer #3 final comments and we are glad we addressed the majority of her/his points raised in the previous revision of our manuscript. Importantly, we also would like to acknowledge her/his commitment in reviewing our work and her/his comments that improved the current version of our manuscript. Below we comment the specific points raised by Reviewer #3 in the current revision:

“They did not address my question of their cell lines being tetraploid and whether each X chromosome is mutated in the same way (point 6)”

We apologize for the misunderstanding regarding our response about the tetraploid nature of S2R+ cells and whether they are mutated in the same way in our CRISPR clones. We stated in point 6 of our previous rebuttal letter:

We apologize for the misunderstanding. As stated in the Methods section:

Line 567:

“Single clones were expanded and an aliquot of cells was used for DNA extraction by phenol-chloroform and PCR genotypification using specific primers spanning the desired deletions (see Table 2). Homozygous clones were expanded and used for subsequent experiments. Mutant clones were further characterized by Sanger Sequencing using primers for genotypification. Electropherograms of breakpoints for all CRISPR mutants generated in this study are shown in Supplementary Fig. 3.”

All our CRISPR results are derived from single clones isolated by serial dilutions from pools of CRISPR mutant cells. We validated the presence of just the mutant band expected from each mutant by PCR in conjunction with the absence of the wild-type band (see for example Figure 1c). **We also determined the identity of the CRISPR mutant by Sanger Sequencing of the mutant band obtained by PCR.**

For each of our CRISPR mutant clones we amplified the mutated region by PCR, cloned into pGEM and sequenced two individual clones per mutant. Since we did not observe differences in the sequence of independent clones containing the mutated fragments we reasoned that the junction resulting after DNA repair due to CRISPR-Cas9 cut was likely the same in the different alleles of S2R+ cells. However, since each cut by the CRISPR-Cas9 system is independent at each allele we cannot rule out the existence of specific indels resulting at each deletion junction at each allele, that can not be detected neither by visual inspection of agarose gels nor in the sequenced clones by Sanger Sequencing.

Regarding the issues raised by Reviewer #3 in this revision we have included the following paragraph in the Methods section of the manuscript:

Single clones were expanded and an aliquot of cells was used for DNA extraction by phenol-chloroform and PCR genotypification using specific primers spanning the desired deletions (see Supplementary Table 2). Homozygous clones were expanded and used for subsequent experiments. Mutant clones were further characterized by Sanger Sequencing using primers for genotypification. Amplified fragments were ligated into pGEM-T Easy and two individual clones were used for Sanger Sequencing per mutant. The sequence of each mutant fragment was the same between clones and Electropherograms of breakpoints for all CRISPR mutants generated in this study are shown in Supplementary Fig. 4. Importantly, S2R+ cells are tetraploid and since each cut by the CRISPR-Cas9 system is independent at each allele we cannot rule out the existence of specific indels resulting at each deletion junction on different alleles, that cannot be detected neither by visual inspection of agarose gels nor in the sequenced clones by Sanger Sequencing.

Finally, regarding the differences between female and male cells we have included a description of the differences between female and male cell lines at *Notch* both in the Results section and in the Discussion:

There are some minor English points that would be good to correct. Here are a few:

line 58- "precluding" instead of "preclude"

Corrected as: "precluding"

line 71: "decrease in TAD boundaries insulation" I would say "decrease in TAD boundary insulation" or "decrease in insulation of TAD boundaries"

Corrected as: "decrease in TAD boundary insulation"

line 72: "On the other hand, depletion of BEAF-32, AP enriched at TAD boundaries, result in minor changes in genome organization" should be "On the other hand, depletion of BEAF-32, an AP enriched at TAD boundaries, results in minor changes in genome organization"

Corrected as: "On the other hand, depletion of BEAF-32, an AP enriched at TAD boundaries, results in minor changes in genome organization"

line 111: "Quality controls of all replicates are of highest standards," I would avoid subject statements like this in the results and simply state that "a minimum of 89% valid pairs per replicate was achieved"

Corrected as: "reaching a minimum of 89% valid pairs per replicate."